# SANA-VIDEO: EFFICIENT VIDEO GENERATION WITH BLOCK LINEAR DIFFUSION TRANSFORMER

**Junsong Chen**[1,2*], **Yuyang Zhao**[2*], **Jincheng Yu**[2*], **Ruihang Chu**[4], **Junyu Chen**[2],

**Shuai Yang**[2], **Xianbang Wang**[3], **Yicheng Pan**[4], **Daquan Zhou**[5], **Huan Ling**[2],

**Haozhe Liu**[6], **Hongwei Yi**[2], **Hao Zhang**[2], **Muyang Li**[3], **Yukang Chen**[2], **Han Cai**[2],

**Sanja Fidler**[2], **Ping Luo**[1], **Song Han**[2,3], **Enze Xie**[2]

[1]**HKU**   [2]**NVIDIA**   [3]**MIT**   [4]**THU**   [5]**PKU**   [6]**KAUST**

Project Page: https://nvlabs.github.io/Sana/Video/

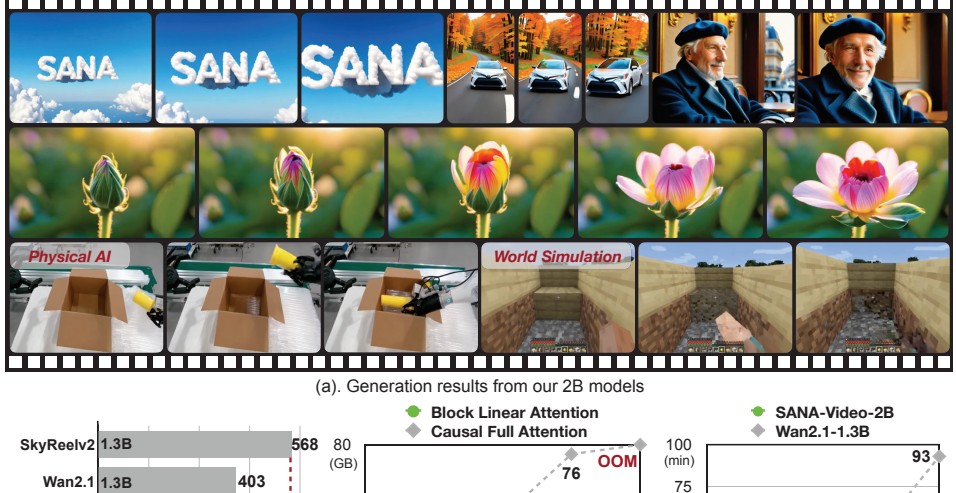

(a). Generation results from our 2B models

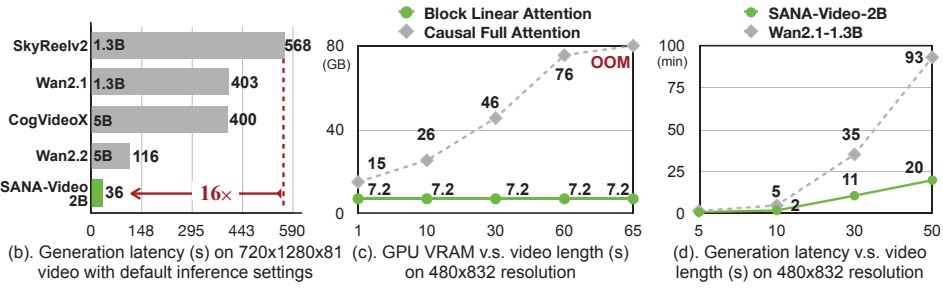

(b). Generation latency (s) on 720x1280x81 video with default inference settings

(c). GPU VRAM v.s. video length (s) on 480x832 resolution

(d). Generation latency v.s. video length (s) on 480x832 resolution

Figure 1: An overview of generated videos and inference latency and memory of SANA-Video. The generation latency is measured under 50 denoising steps. Linear attention is more efficient for video generation and our block linear attention maintains a fixed memory requirement for long videos.

## ABSTRACT

We introduce SANA-Video, a small diffusion model that can efficiently generate videos up to 720×1280 resolution and minute-length duration. SANA-Video synthesizes high-resolution, high-quality and long videos with strong text-video alignment at a remarkably fast speed, deployable on RTX 5090 GPU. Two core designs ensure our efficient, effective and long video generation: (1) **Linear DiT:** We leverage linear attention as the core operation, which is more efficient than vanilla attention given the large number of tokens processed in video generation. (2) **Constant-Memory KV cache for Block Linear Attention:** we design block-wise autoregressive approach for long video generation by employing a constant-memory state, derived from the cumulative properties of linear attention. This KV cache provides the Linear DiT with global context at a fixed memory cost, eliminating the need for a traditional KV cache and enabling efficient, minute-long video generation. In addition, we explore effective data filters and model training strategies, narrowing the training cost to **12 days on 64 H100 GPUs**, which is only **1%** of the cost of MovieGen. Given its low cost, SANA-Video achieves competitive performance compared to modern state-of-the-art small diffusion models (*e.g.*, Wan 2.1-1.3B and SkyReel-V2-1.3B) while being **16×** faster in measured la-

tency. Moreover, SANA-Video can be deployed on RTX 5090 GPUs with NVFP4 precision, accelerating the inference speed of generating a 5-second 720p video from 71s to 29s (**2.4**× speedup). In summary, SANA-Video enables low-cost, high-quality video generation. Code and model will be publicly released.

# 1 INTRODUCTION

Video generation is currently a highly active field, fueling applications that range from creative content production and digital live streaming to virtual product displays. Recent large-scale models from industry labs, such as Veo3 (DeepMind, 2025), Kling (Kuaishou, 2024), Wan (Wang et al., 2025a) and Seedance (Gao et al., 2025), have demonstrated remarkable performance in generating high-fidelity video content. However, this quality comes at the cost of immense computational complexity. Video generation is an exceptionally token-extensive task; for instance, producing a single 5-second video at 720p resolution with a model like Wan 14B (Wang et al., 2025a) requires to process over 75,000 tokens, taking 32 minutes on a H100 GPU. This sheer volume of data leads to prohibitive training costs and extremely slow generation speeds, rendering these powerful models impractical for widespread research and application. Even with large cost, generating long video (>10 s) is hard to realize with these large models due to the full-sequence processing operation. Recent works (*e.g.*, MAGI-1 (Teng et al., 2025) and SkyReelv2 (Chen et al., 2025a)) explores the long video generation but the efficiency is strictly constrained by the vanilla attention and KV cache. Given these challenges, a pivotal question arises: *Can we develop a high-quality and high-resolution video generator that is computationally efficient and runs very fast on both cloud and edge devices?*

This paper proposes SANA-Video, a small diffusion model designed for both efficient training and rapid inference without compromising output quality. In stark contrast to the massive resource requirements of contemporary models, SANA-Video's training is remarkably cost-effective, requiring only **64 NVIDIA H100 GPUs for 12 days**, which represents as little as 1% of the training cost of MovieGen (Polyak et al., 2024) and 10% of that of OpenSora (Zheng et al., 2024). This efficiency extends to inference, where SANA-Video can generate a 5-second, 720p video in just 36 seconds on a NVIDIA H100 GPU. By drastically reducing the computational barrier, SANA-Video makes high-quality video generation more accessible and practical for a broader range of users and systems. The improvements mainly lie in three key components.

**Linear DiT.** We extend SANA (Xie et al., 2025a) linear DiT design to the video domain, addressing the significant computational bottleneck of traditional self-attention ($O(N^2)$), as shown in Fig. 1(d). By replacing all attention modules with our efficient linear attention, we reduce complexity to $O(N)$, which is crucial for high-resolution video generation and leads to a 4× acceleration on 720p video. To enhance our model for video, we make two key improvements. We first integrate Rotary Position Embeddings (RoPE) (Su et al., 2024) to improve long-context modeling. In Sec. 3.2, we detail our exploration of the optimal placement for RoPE and how we address the training instability it can introduce. Additionally, we introduce a 1D temporal convolution to the Mix-FFN via a shortcut connection. This design allows us to effectively leverage pre-trained image models and efficiently adapt them for video generation by aggregating temporal features.

**Block Linear Attention with KV Cache.** The success of SANA-Video in long video generation is mainly inspired by the attribute of causal linear attention (Katharopoulos et al., 2020). Based on our reformulation of the causal linear attention operation, we reduce the KV cache to a small and fixed memory, along with a fixed computational cost for each new token. This natively supports long-context operations. Based on the block linear attention module, we introduce a two-stage autoregressive model continue-training paradigm, including autoregressive block training with monotonically increasing SNR sampler and the improved self-forcing specially for our long context attention operation, leading to efficient, long, and high quality video generation.

**Efficient Data Filter and Training.** The low training cost is mainly attribute to three aspects: the powerful pre-trained text-to-image (T2I) model, efficient data filtering, and the efficient training strategy. First, SANA-Video is continue pre-trained from SANA (Xie et al., 2025a;b)-1.6B T2I model with the modification for spatio-temporal modeling (Sec. 3.2). Second, we collect data from diverse data source and design specific data filtering criterion for each data source. In addition, a strong VLM (Bai et al., 2025a) serves as our video captioner, producing highly detailed captions (80-100 words), including subject category, color, appearance, actions, expressions, surrounding

environment, camera angles, etc. Third, with the high-quality video-text pairs, we train SANA-Video in multiple stages from low resolution to high resolution and finally leverage human preferred data for SFT, ensuring the model can efficiently learn the motion and aesthetic appearance.

In conclusion, our model achieves a latency that is over $13\times$ faster than the state-of-the-art Wan2.1 for 720p video generation (Fig. 1(b)), while delivering competitive results across many benchmarks. Additionally, we quantize and deploy our SANA-Video on RTX 5090 GPUs with the NVFP4 precision, where it takes just 29 seconds to generate a 5s 720p video. We hope our model can be efficiently used by everyday users, providing a powerful foundation model for fast video generation.

## 2 PRELIMINARIES

### 2.1 VIDEO DIFFUSION MODEL

Following SANA (Xie et al., 2025a), we use Rectified Flows (RFs) (Esser et al., 2024) with SNR sampler as the training objective in Eq. 1. Here, $c$ is the conditional embedding, $\theta$ is the model weights, and $u(x^t \mid t, c; \theta)$ denotes the output velocity predicted by the diffusion model. $v(x)$ is the target velocity. In this paper, our SANA-Video is a unified framework for Text-to-Image (T2I), Text-to-Video (T2V), and Image-to-Video (I2V) generation by varying condition embeddings. Specifically, for T2I and T2V, $c$ is the text prompt and $x$ is the image or video. For I2V, we use first frame and text prompt as condition $c$. By setting the noise of the first frame to zero, SANA-Video can realize I2V without any model modification. Therefore, the joint training of T2I, T2V, and I2V makes SANA-Video a unified framework that can perform all tasks with a single model.

$$\mathbb{E}_{c,t,x^0} \left\| u\big(x^t \mid t, c; \theta\big) - v(x) \right\|^2. \tag{1}$$

### 2.2 AUTOREGRESSIVE LONG VIDEO GENERATION

Autoregressive diffusion models combine a token/block-wise autoregressive chain-rule decomposition with denoising diffusion models, emerging as a promising direction for long sequence generation like language (Arriola et al., 2025) and video generation (Yin et al., 2025; Chen et al., 2025a; Huang et al., 2025). Specifically, for a sequence of $N$ blocks $x_{1:N} = (x_1, x_2, \ldots, x_N)$, the generation process is a product of block distribution using the chain rule $p(x_{1:N}) = \prod_{i=1}^{N} p(x_i|x_{j<i})$, with each block distribution $p(x_i|x_{j<i})$ modeled using a diffusion process (Eq. 1). This approach leverages the strengths of both autoregressive models and diffusion models to capture sequential dependencies and enable block-wise, high-quality generation.

## 3 SANA-VIDEO

Scaling video generation to higher resolutions and longer sequences dramatically increases the number of tokens, making the $O(N^2)$ complexity of self-attention a major bottleneck in computation, speed, and memory. This underscores the need for efficient linear attention in video generation. Building upon SANA Linear DiT (Xie et al., 2025a), we introduce Linear Video DiT (Fig. 2(a)) for video generation by integrating two key components: Rotary Position Embeddings (RoPE) and a temporal 1D convolution within the Mix-FFN. These designs keep SANA's macro architecture as well as additional temporal modeling (Fig. 2(b)), allowing us to leverage a pre-trained image model and efficiently adapting it into a powerful video model through continuous pre-training. In addition to the short video generation, we introduce block linear attention module for efficient long video generation. With the re-formulation of linear attention, the block linear attention module and causal Mix-FFN keep a constant-memory KV cache and linear computational cost for long video. Based on this KV cache, we design two stage post-training paradigm to unlock the infinite length generation ability, leading to a high-quality and efficient long video generation model.

### 3.1 TRAINING STRATEGY

**Stage1: VAE adaptation on text-to-image (T2I).** Training video DiT models from scratch is resource-intensive due to the mismatch between image and video VAEs. We address this by first efficiently adapting existing T2I models to new video VAEs. Specifically, We leverage different video VAEs in generating videos of different resolution. For 480P videos, VAEs with high-compression

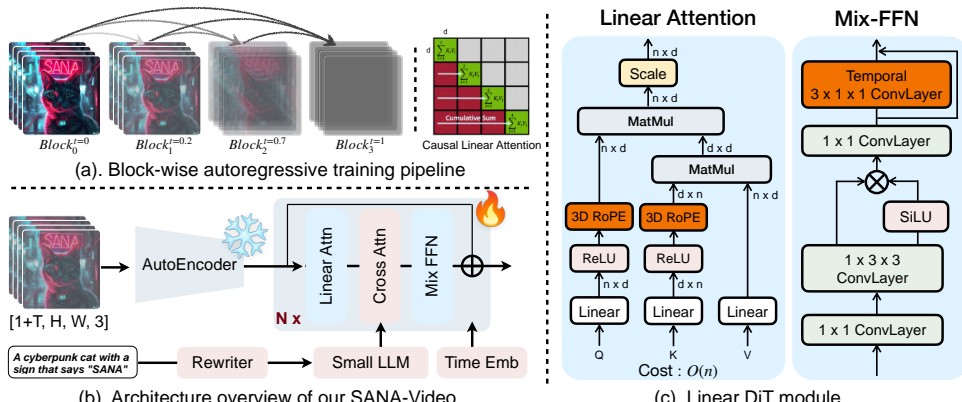

Figure 2: **Overview of SANA-Video.** Fig.(a) A high-level block-wise autoregressive training pipeline based on our block causal KV cache. (Details in Sec. 3.3). Fig.(b) Our model pipeline, containing an Autoencoder, Re-writer, Linear DiT, and a text encoder. Fig.(c) The detailed design of the added 3D RoPE in linear attention and the temporal convolution in our Linear DiT's Mix-FFN.

ratio limits the overall performance, and thus we adopt Wan-VAE (Wang et al., 2025a). For 720P high-resolution video, we introduce our video VAE, DCAE-V, which provides a higher compression ratio for more efficient generation (details in Sec. 3.5). The adaptation of both VAEs is highly efficient, converging within 5-10k training steps, further demonstrating the strong generalization ability of our Linear DiT.

**Stage2: Continue pre-training from T2I model.** Initializing video Linear DiT from a pre-trained T2I model (Xie et al., 2025a) is an efficient and effective way to leverage the well-learned visual and textual semantic knowledge. Therefore, we initialize our SANA-Video with a model adapted from the first stage and introduce additional designs to model long-context and motion information (Sec. 3.2). The additional temporal designs are tailor-made for linear attention, improving the locality of attention operation. The newly added layers are zero-initialized with skip connection, which minimizes their influence on the pre-trained weights during early training. After this identity initialization, SANA-Video is trained in a coarse-to-fine manner. It first trains on low-resolution, short videos (*e.g.*, 192P 2.5 seconds) before moving to higher resolution, longer videos (*e.g.*, 480P 5 seconds) with different data filtering criteria (Appendix E). This coarse-to-fine approach efficiently encourages SANA-Video to fast learn dynamic information with abundant data and then refine details using less, but higher-quality, data.

**Stage3: Autoregressive block training.** The continued pre-training makes SANA-Video an efficient small diffusion model, primarily for high-resolution 5-second video generation. To enable the generation of much longer videos, we analyze the attributes of linear attention in Sec. 3.3 and propose a constant-memory block KV cache for autoregressive generation. Building on this design, we conduct autoregressive block training in two steps: we first train the autoregressive module and then address exposure bias with our improved self-forcing block training (Sec. 3.4). This process results in a high-quality, efficient model for long video generation.

## 3.2 EFFICIENT LINEAR DIT PRETRAINING

SANA-Video adopts the SANA (Xie et al., 2025a) as the base architecture and innovatively tailors the Linear Diffusion Transformer blocks to handle the unique challenges of T2V tasks, as depicted in Fig. 2. Several dedicated designs are proposed as follows:

**Linear Attention in Video DiT.** Our work extends the SANA (Xie et al., 2025a) architecture by integrating Rotary Position Embeddings (RoPE) (Su et al., 2024) into its efficient ReLU ($\phi$) linear attention blocks. This integration is crucial for enhancing the model's ability to handle the sequential and spatial relationships in high-quality video generation. The core of our design lies in applying RoPE after the ReLU activation, specifically as RoPE(ReLU($x$)), as shown in Fig. 2. This order is critical because it prevents the ReLU kernel from filtering out the positional information encoded

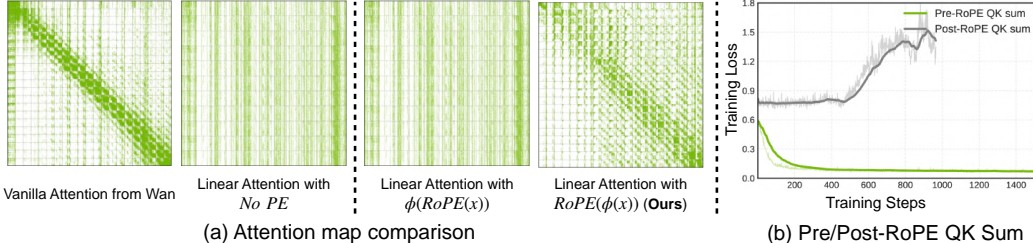

(a) Attention map comparison    (b) Pre/Post-RoPE QK Sum

Figure 3: **Analysis of Linear Attention with RoPE.** (a) Visual comparison of attention maps. First two plots compare vanilla softmax attention (Wan) to our linear attention without positional encoding. The latter two plots show our method's effect: applying RoPE after the ReLU kernel results in a sparser, more localized attention pattern. (b) Training loss for the QK sum (Eq. 2 denominator). Removing RoPE from the denominator (green line) ensures training stability, as discussed in Sec. 3.2.

by RoPE. As Fig. 3 shows, this design results in attention maps with a clear focus on local regions, which is essential for capturing fine-grained video details. However, applying RoPE directly to queries and keys (as in vanilla attention) can make the linear attention mechanism numerically unstable (Su, 2021) due to the difference between softmax and ReLU similarity functions. The RoPE transformation can change the non-negative nature of the ReLU output, potentially causing the denominator in the standard linear attention formula (Eq. 2) to become zero. To solve this, we modify the calculation: while the numerator includes RoPE on the queries and keys, we remove RoPE from either the key or the query in the denominator. This ensures the denominator remains positive, guaranteeing training stability (Fig. 3 (b)) while still benefiting from positional encoding.

$$O_i = \frac{\text{RoPE}(\phi(Q_i))\left(\sum_{j=1}^{N} \text{RoPE}(\phi(K_j))^T V_j\right)}{\text{RoPE}(\phi(Q_i))\left(\sum_{j=1}^{N} \text{RoPE}(\phi(K_j))^T\right)} \implies \frac{\text{RoPE}(\phi(Q_i))\left(\sum_{j=1}^{N} \text{RoPE}(\phi(K_j))^T V_j\right)}{\phi(Q_i)\left(\sum_{j=1}^{N} \phi(K_j)^T\right)}, \tag{2}$$

where $O_i$, $Q_i$, $K_i$ and $V_i$ denote the output, query, key and value of the $i$th token.

**Mix-FFN with Spatial-Temporal Mixture.** As shown in Fig. 3, we compare the linear attention map in SANA-Video with the softmax attention map in Wan2.1 (Wang et al., 2025a). We observe that linear attention is much denser and less focused on local details compared to softmax attention. SANA (Xie et al., 2025a) ameliorates the locality problem in image generation with the convolution in Mix-FFN. Building upon the Mix-FFN, we enhance it with a temporal 1D convolution. The temporal convolution with a shortcut connection is appended to the end of the block (Fig. 2(b)), enabling seamless temporal feature aggregation while preserving initialization. The module helps capture local relationships along the temporal axis, resulting in better motion continuity and consistency in generated videos. As evidenced in our ablation study (Fig. 5(a)), this addition leads to a significantly lower training loss and improved motion performance.

## 3.3 BLOCK LINEAR ATTENTION

This section outlines key components enabling efficient long-video generation. Inspired by the inherent attribute of causal linear attention (Katharopoulos et al., 2020), we explore the **constant-memory global KV cache** in our block linear attention module, which supports long-context attention with small, fixed GPU memory. Based on this module, we introduce a two-stage autoregressive model continue training paradigm: autoregressive block training with a monotonically increasing SNR sampler and an improved self-forcing method for our long-context attention.

### 3.3.1 BLOCK LINEAR ATTENTION WITH KV CACHE

Table 1: For a sequence with $N$ tokens $\in \mathbb{R}^{1 \times D}$, memory and compute costs are compared among three attention types. Causal linear attention shows best efficiency while maintains global memory.

| Metric | Causal Full Attention | Causal Local Attention | Causal Linear Attention |
|---|---|---|---|
| Memory | $O(N \times D)$ | $O(W \times D)$ | $O(D^2)$ |
| Comp. Cost ($N$-th token) | $O(N \times D)$ | $O(W \times D)$ | $O(D^2)$ |
| Comp. Cost ($N$ tokens) | $O(N^2 \times D)$ | $O(N \times W \times D)$ | $O(N \times D^2)$ |

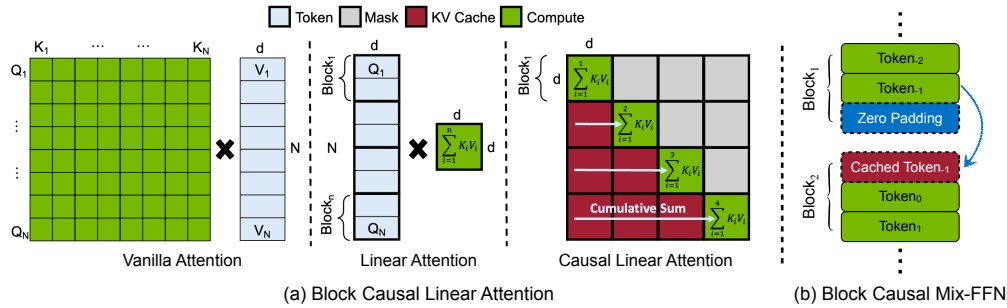

Figure 4: **Overview of Block Linear Attention.** (a) We compare the attention compute mechanism among vanilla attention, linear attention and causal linear attention. (b) The illustration of block causal Mix-FFN in processing the adjacent blocks.

**Limitation of Causal Vanilla Attention.** In view of the training objective (Eq. 1), block-wise causal attention is required to implement autoregressive generation. Recent works (Huang et al., 2025; Chen et al., 2025a; Teng et al., 2025) use a combination of full attention within a block and causal attention to previous blocks. To reduce computational costs, they leverage KV cache, which is effective but comes with memory overhead. For each new token $\in \mathbb{R}^{1 \times D}$ with $N$ cached conditional tokens, it requires $O(N \times D)$ memory to store the cache and $O(N \times D)$ FLOPs for the attention computation. However, since the computational and memory costs grow linearly, these methods (Huang et al., 2025; Chen et al., 2025a; Teng et al., 2025) often restrict the attention window to a local scope during long video generation. While this maintains a stable cost, it comes at the expense of losing global-context information.

**KV cache in Block Linear Attention.** In contrast to the dramatically increased computational and memory cost in causal vanilla attention, linear attention (Katharopoulos et al., 2020) has significant efficiency advantage, naturally supporting long video generation with **global attention** while maintaining **constant memory**. Consider the causal attention setting, linear attention (Eq. 2) output for the $i$th token can be re-formulated as:

$$O_i = \frac{\phi(Q_i)\left(\sum_{j=1}^{i}\phi(K_j)^T V_j\right)}{\phi(Q_i)\left(\sum_{j=1}^{i}\phi(K_j)^T\right)} = \frac{\phi(Q_i)\left(\sum_{j=1}^{i-1}\phi(K_j)^T V_j + \phi(K_i)^T V_i\right)}{\phi(Q_i)\left(\sum_{j=1}^{i-1}\phi(K_j)^T + \phi(K_i)^T\right)} = \frac{\phi(Q_i)\left(\sum_{j=1}^{i-1} S_j + S_i\right)}{\phi(Q_i)\left(\sum_{j=1}^{i-1}\phi(K_j)^T + \phi(K_i)^T\right)}, \tag{3}$$

where $S_j = \phi(K_j)^T V_j$ denotes the attention state for the $j$th token. We omit RoPE here for simplicity. Obviously, as long as the cumulative sum of state $\sum_{j=1}^{i-1} S_j$ and the cumulative sum of keys $\sum_{j=1}^{i-1} \phi(K_j)^T$ are stored, only the attention state for the $i$th token $S_i \in \mathbb{R}^{D \times D}$ is required to compute. Therefore, the memory cost is only $\sum_{j=1}^{i-1} S_j \in \mathbb{R}^{D \times D}$ and $\sum_{j=1}^{i-1} \phi(K_j)^T \in \mathbb{R}^{D \times 1}$, taking $O(D^2)$ in total, and the computational cost is only $O(D^2)$. In Table 1 and Fig. 4(a), we compare the memory and computational cost among causal full attention, causal local attention and our causal linear attention. Since $N > W >> D$, causal linear attention achieves the best efficiency and can still maintain global memory in long video generation.

**Block Causal Mix-FFN.** In addition to linear attention, our proposed temporal-spatial Mix-FFN enhances locality using convolutional layers. To support long video generation, this module must also operate causally. We ensure causal processing during both training and inference with two operations, as illustrated in Fig. 4(b). First, to prevent information leakage from subsequent blocks during training, we append an all-zero token ('Zero Padding' $\in \mathbb{R}^{1 \times HW \times D}$) to the end of each block $\in \mathbb{R}^{T \times HW \times D}$. Second, our causal temporal convolution (kernel size 3) requires the last frame of the preceding block. We address this by caching the last token of each block ('$Token_{-1}$' $\in \mathbb{R}^{1 \times HW \times D}$) and prepending it to the next. Overall, our causal linear DiT module keeps a fixed memory cache, containing cumulative sum of attention states and keys from all previous frames for attention, along with the last frame of the previous block for Mix-FFN.

### 3.4 LONGSANA

**Autoregressive Block Training.** The continue training of the autoregressive SANA-Video variant, *i.e.* LongSANA, begins with the pre-trained 5s SANA-Video model. To align with the pre-trained

model's distribution, we propose a monotonically increasing SNR sampler. Specifically, we randomly select a block and sample a timestep for it with the SNR sampler (Esser et al., 2024). Then the timesteps for the remaining blocks are sampled via propagated probability (Sun et al., 2025), ensuring all the timesteps are monotonically increasing, *i.e.*, later blocks have larger timestep than early blocks. This proposed timestep sampler offers two key advantages. First, the monotonically increasing timesteps have a much smaller sampling space than random timesteps, which results in faster convergence and better performance. Second, applying the SNR sampler to a randomly selected block guarantees that every block is trained with sufficient information.

**Addressing Exposure Bias with Long Training.** However, monotonically increasing SNR sampler cannot address a severe problem in autoregressive generation, *i.e.*, exposure bias, where condition blocks are ground truth during training but are generated content during inference, leading to error accumulation and limiting performance in long video generation. Self-Forcing (Huang et al., 2025) aims to address this issue in a vanilla attention DiT model with autoregressive rollout. Limited by the increasing VRAM requirement of causal vanilla attention (Fig. 1(c) and Table 1), Self-Forcing uses local attention within a designed window size. Consequently, it sets the length of self-generated content to be the same as the pre-trained model (*i.e.*, 5s). Later on, LongLive (Yang et al., 2025a) explores streaming long training on 1 minute video, but it still limits to the local attention with sink due to the complexity of full attention. In contrast to the full attention, the block linear attention in LongSANA supports a long-context global KV cache with a small and constant GPU memory. This allows us to further extend LongLive with global attention when self-generating a much longer video (*e.g.*, 1 min), which better aligns the conditioning signals between training and inference and keeps better temporal consistency. The inference details are illustrated in Alg. 1.

## 3.5 DEEP COMPRESSION VIDEO AUTOENCODER

SANA-Video achieves high efficiency and quality for 480P video generation using Wan-VAE. However, even with our efficient linear attention, the generation speed for 720P videos is $2.3\times$ slower. This efficiency drop is even more severe for full attention DiT models ($4\times$ for Wan 2.1 1.3B), inspiring us to explore a more efficient VAE that can compress more tokens. We fine-tune DCAE (Chen et al., 2024c) into DCAE-V, with a spatial down-sampling factor of $F = 32$, a temporal factor of $T = 4$, and channels $C = 32$. The number of latent channels aligns with our pre-trained T2I model, enabling fast adaptation from an image to a video model in the same latent space.

The concurrent Wan2.2-5B model also achieves 32 times spatial compression, by combining a VAE with a spatial down-sampling factor of 16 and a patch embedding compression of 2. The advantages of DCAE-V over Wan2.2-VAE are twofold. First, DCAE-V's 32 latent channels align with our pre-trained T2I model, which improves convergence speed. Second, to achieve the same compression ratio, Wan2.2-VAE would require the model to predict a much larger latent dimension (192 vs. 32 in DCAE-V), a task that is difficult for a small diffusion model (Details in Appendix D.1). As shown in Table 3, DCAE-V exhibits reconstruction performance comparable to other state-of-the-art VAEs like Wan2.1 (Wang et al., 2025a), Wan2.2 (Wang et al., 2025a), and LTX-Video (HaCohen et al., 2024). This high compression allows our model to achieve performance on par with much larger models (*e.g.*, Wan2.1-14B and Wan2.2-5B) while demonstrating significant acceleration, as shown in Table 2. Specifically, SANA-Video can generate a 720P 5s video within just 36 seconds, which is a $53\times$ acceleration over Wan2.1-14B. When compared to Wan2.2-5B, which shares the same compression ratio as ours, SANA-Video achieves a $3.2\times$ acceleration.

Table 2: Latency on H100 GPU and VBench evaluation on $720\times1280\times81$ resolution videos.

| Models | Latency(s) | Total ↑ | Quality ↑ | Semantic ↑ |
|---|---|---|---|---|
| Wan-2.1-14B | 1897 | 83.73 | 85.77 | 75.58 |
| Wan-2.1-1.3B | 400 | 83.38 | 85.67 | 74.22 |
| Wan-2.2-5B | 116 | 83.28 | 85.03 | 76.28 |
| **SANA-Video-2B** | 36 | 84.05 | 84.63 | 81.73 |

Table 3: Reconstruction capability of different Autoencoders on Panda-70M 192p resolution.

| Autoencoder | Ratio | PSNR ↑ | SSIM ↑ | LPIPS ↓ |
|---|---|---|---|---|
| F8T4C16 (Wan2.1-VAE) | 16 | 34.41 | 0.95 | 0.01 |
| F16T4C48 (Wan2.2-VAE) | 21 | 35.61 | 0.96 | 0.01 |
| F32T8C128 (LTX-VAE) | 64 | 32.26 | 0.93 | 0.04 |
| F32T4C32 (**Our DCAE-V**) | 128 | 33.25 | 0.94 | 0.03 |

Table 4: **Comprehensive comparison of our method with SOTA approaches in efficiency and performance on VBench.** The speed is tested on one H100 GPU with BF16 Precision. Latency: Measured with a batch size of 1, on a 480×832×81 video, using the model's default inference steps for a fair comparison. We highlight the **best**, second best, and *third best* entries.

| Methods | Latency (s) | Speedup | #Params (B) | Total | Quality | Semantic / I2V |
|---------|-------------|---------|-------------|-------|---------|----------------|
| *Text-to-Video* | | | | | | |
| MAGI-1 (Teng et al., 2025) | 435 | 1.1× | 4.5B | 79.18 | 82.04 | 67.74 |
| Step-Video (Ma et al., 2025) | 246 | 2.0× | 30B | 81.83 | 84.46 | 71.28 |
| CogVideoX1.5 (Yang et al., 2024) | 111 | 4.4× | 5B | 82.17 | 82.78 | *79.76* |
| SkyReels-V2 (Chen et al., 2025a) | 132 | 3.7× | 1.3B | 82.67 | 84.70 | 74.53 |
| Open-Sora-2.0 (Peng et al., 2025) | 465 | 1.0× | 14B | **84.34** | 85.40 | 80.12 |
| Wan2.1-14B (Wang et al., 2025a) | 484 | 1.0× | 14B | *83.69* | **85.59** | 76.11 |
| Wan2.1-1.3B (Wang et al., 2025a) | 103 | 4.7× | 1.3B | 83.31 | *85.23* | 75.65 |
| **SANA-Video** | 60 | 8.0× | 2B | 83.71 | 84.35 | **81.35** |
| *Image-to-Video* | | | | | | |
| MAGI-1 (Teng et al., 2025) | 435 | 1.1× | 4.5B | **89.28** | **82.44** | 96.12 |
| Step-Video-TI2V (Ma et al., 2025) | 246 | 2.0× | 30B | 88.36 | 81.22 | *95.50* |
| CogVideoX-5b-I2V (Yang et al., 2024) | 111 | 4.4× | 5B | 86.70 | 78.61 | 94.79 |
| HunyuanVideo-I2V (Kong et al., 2024) | 210 | 2.3× | 13B | 86.82 | 78.54 | 95.10 |
| Wan2.1-14B (Wang et al., 2025a) | 493 | 1.0× | 14B | 86.86 | *80.82* | 92.90 |
| **SANA-Video** | 60 | 8.2× | 2B | *88.02* | 79.65 | **96.40** |

## 4 EXPERIMENTS

### 4.1 IMPLEMENTATION DETAILS.

**Pipeline Settings.** For DiT model, to best utilize the pre-trained text-to-image model SANA (Xie et al., 2025a), our SANA-Video-2B is almost identical to those of the original SANA (Xie et al., 2025a), including the diffusion transformer model and small decoder-only text encoder. For 480P videos, we leverage a Wan2.1-VAE (Wang et al., 2025a) autoencoder. For 720P high-resolution video generation, we fine-tune the DCAE (Chen et al., 2024c) into a video deep compression autoencoder (DCAE-V) to facilitate more efficient training and inference. Our final model is trained on 64 H100 GPUs for approximately 12 days. Details are in Appendix C.1.

### 4.2 PERFORMANCE COMPARISON AND ANALYSIS

The comprehensive efficiency and performance comparison among SANA-Video with state-of-the-art is illustrated in Table 4. We adopt VBench (Zhang et al., 2024) as the performance evaluation metric and the generation latency of a 480P 81-frame video as efficiency metric. As shown in Table 4, SANA-Video exhibits remarkable latency of 60 seconds, marking it the fastest model compared. This translates to a throughput that is 7.2× faster than MAGI-1 and over 4× faster than Step-Video. In terms of comparison, SANA-Video achieves a Total Score of 83.71 on text-to-video generation, comparable with large model Open-Sora-2.0 (14B) and outperforming Wan2.1 (1.3B). In addition, SANA-Video achieves 88.02 Total Score on image-to-video generation, outperformance large DiT models Wan2.1 (14B) and HunyuanVideo-I2V (11B). Furthermore, SANA-Video achieves the best semantic / I2V score across all the methods, demonstrating strong vision-text semantic alignment.

### 4.3 ABLATION STUDIES

We then conduct ablation studies on the crucial architectural modifications discussed in Sec. 3.2. As shown in Fig. 5, we provide training loss curves and latency profiles on H100 GPUs.

**Linear Attention Module.** We incorporate three key designs to enhance our linear attention model. First, we integrate 3D RoPE to focus linear attention on local features (Fig. 3). This improves performance, as evidenced by a significantly lower training loss (Fig. 5(a)). Second, to address differences between linear and vanilla attention, we introduce a Spatial-Temporal Mix-FFN module. Its training loss curve (Fig. 5(b)) demonstrates that a 1D temporal convolution layer significantly enhances

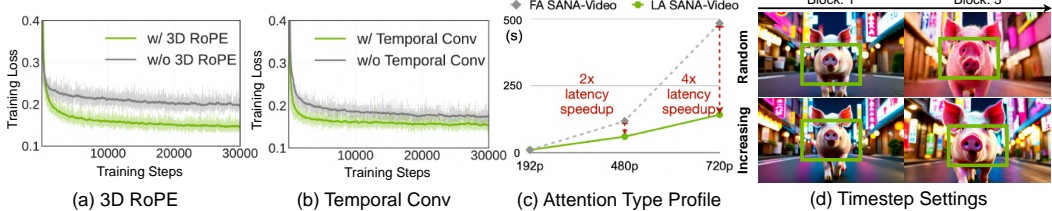

(a) 3D RoPE     (b) Temporal Conv     (c) Attention Type Profile     (d) Timestep Settings

Figure 5: **SANA-Video configuration ablation studies.** (a) Training loss curves with and without 3D RoPE. (b) Training loss curves with and without temporal 2D Convolution. (c) Latency comparison of SANA-Video between linear and full attention. (d) Comparison of monotonically increasing versus random timestep sampling in autoregressive block training.

performance. Finally, our linear attention design provides a significant efficiency advantage. As Fig. 5(c) shows, our model's latency becomes lower at higher resolutions, achieving a $2\times$ speedup at 480P and $4\times$ at 720P, proving its superior efficiency for high-resolution video generation.

**Monotonically Increasing SNR Sampler.** We compare the proposed monotonically increasing SNR sampler with random timestep sampling in the autoregressive block training. As shown in Fig. 5(d) (two columns are from different blocks), monotonically increasing SNR sampler achieves better quality and more consistency across blocks.

We also provide quantitative ablations studies for 3D RoPE, temporal conv and monotonically increasing SNR sampler in Table 5. All the three design is crucial for achieving our high-quality, efficient and long video generation.

Table 5: **Quantitative ablation studies on VBench.**

| Model | Total Score ↑ | Quality Score ↑ | Semantic Score ↑ |
|---|---|---|---|
| w/o Temporal Conv | 80.94 | 82.63 | 74.18 |
| w/ Temporal Conv | 81.71 | 83.10 | 76.15 |
| w/o 3D RoPE | 81.19 | 82.68 | 75.22 |
| w/ 3D RoPE | 82.79 | 83.89 | 78.38 |
| Random Steps | 82.00 | 83.13 | 77.51 |
| Increasing Steps | 83.70 | 84.43 | 80.78 |

**Long Video Generation.** We compare SANA-Video with previous autoregressive video generation methods on VBench, as shown in Table 6. SANA-Video achieves comparable performance with Self-Forcing (Huang et al., 2025) while outperforming SkyReel-V2 (Chen et al., 2025a) and CausVid (Yin et al., 2025).

Table 6: Comparison of autoregressive video generation methods on VBench.

| Model | Total Score ↑ | Quality Score ↑ | Semantic Score ↑ |
|---|---|---|---|
| CausVid | 81.20 | 84.05 | 69.80 |
| SkyReels-V2 | 82.67 | 84.70 | 74.53 |
| Self-Forcing | 84.31 | 85.07 | 81.28 |
| **SANA-Video** | 83.70 | 84.43 | 80.78 |

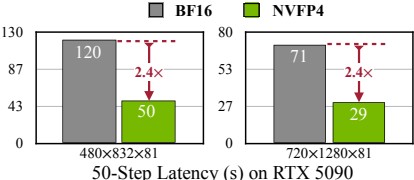

Figure 6: Latency comparison of our model on BF16 and NVFP4 precision.

# 5 APPLICATIONS AND DEPLOYMENT

As a pre-training model, SANA-Video can be easily extended to multiple applications of video generation. First, we adapt SANA-Video to several world model applications (Fig. 1 and Appendix F): embodied AI, autonomous driving and game generation. (Details are in Appendix F). Second, we quantize our model to NVFP4 for efficient inference.

**On-Device Deployment with 4-Bit Quantization.** To facilitate efficient edge deployment, we quantize SANA-Video from BF16 to NVFP4 format using SVDQuant (Li et al., 2024a). To balance

efficiency and fidelity, we selectively quantize the following layers: the QKV and output projections in self-attention, the query and output projections in cross-attention, and the 1x1 convolutions in feed-forward layers. Other components (normalization layers, temporal convolutions, and KV projections in cross-attention) are kept at higher precision to preserve semantic quality and prevent compounding errors. As shown in Fig. 6, this strategy reduces the end-to-end generation time for a 720p 5-second video from **71 s to 29 s** on a single RTX 5090 GPU, achieving a **2.4× latency speedup** while maintaining a quality indistinguishable from the BF16 baseline.

## 6 RELATED WORKS

Video generation has advanced rapidly with diffusion models, evolving from text-to-image extensions with temporal layers (Singer et al., 2022; Wu et al., 2023b) to latent video diffusion (Blattmann et al., 2023a), temporal VAEs, and large-scale diffusion transformers such as Sora (Brooks et al., 2024). Parallel work investigates hybrid autoregressive–diffusion strategies to endow visual models with long-term planning, including block-wise and token-level hybrids (Li et al., 2024b; Hu et al., 2024; Deng et al., 2024), later extended to video with temporal planners (Liu et al., 2024a) and diffusion forcing (Chen et al., 2025a; 2024a). An alternative line of research targets efficiency in training and inference through architectural design. Approaches include linear (Xie et al., 2025a) and state-space attention mechanisms (Liu et al., 2024b) for image generation, as well as mamba-based designs (Wang et al., 2025b; Gao et al., 2024) and spatio-temporal factorization (Chen et al., 2023a; Ho et al., 2022; Wang et al., 2025c; Singer et al., 2022) for video generation. Collectively, these methods highlight the ongoing push toward scalable, efficient, and general-purpose video diffusion models. A more comprehensive related work section is provided in the Appendix B.

## 7 CONCLUSION

In this paper, we introduce SANA-Video, a small diffusion model that can efficiently generate high resolution, high-quality and long videos at a remarkably fast speed and a low hardward requirement. The significance of SANA-Video lies in the following improvements: linear attention as the core operation, leading to remarkable efficiency improvement in token-extensive video generation task; block linear attention with costant-memory KV cache, supporting minute-long video generation with a fixed memory cost; effective data filters and model training strategies, narrowing the training cost to **12 days on 64 H100 GPUs**. With such a small cost, SANA-Video showcases **16× faster** latency but competitive performance with modern state-of-the-art small diffusion models.

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

## A    LLM USAGE

Our use of large language models (LLMs) was limited to editorial assistance to improve the clarity and readability of this manuscript. Specifically, these tools were used to refine grammar and phrasing, enhance the logical flow between sections, and condense overly verbose passages for conciseness. Crucially, all original research ideas, experimental designs, and data analyses were conceived and executed by the authors; the LLM did not contribute to any scientific or methodological content.

## B    FULL RELATED WORK

### B.1    VIDEO DIFFUSION MODEL

Video generation has become a rapidly growing focus in generative AI. Modern approaches typically use a VAE to compress videos into a latent space, where a diffusion model—conditioned on text, images, or both—learns to generate content. Early studies, such as Make-A-Video (Singer et al., 2022), PYoCo (Ge et al., 2023) and Tune-A-Video (Wu et al., 2023b), adapted text-to-image models with additional temporal layers to enable video generation. Works like, MagicVideo (Zhou et al., 2022), SVD (Blattmann et al., 2023a) and Latent Video Diffusion (Blattmann et al., 2023b) played pioneering roles in scaling latent diffusion approaches. However, the limited compression rate of VAEs has hindered their ability to generalize to long video sequences. A major breakthrough came with Sora (Brooks et al., 2024), which introduced a temporal VAE to compress temporal dimensions alongside spatial ones, while adopting a transformer-based backbone (Peebles & Xie, 2023) at scale. Recent efforts have pushed this framework further. For instance, Wan 2.2 (Wang et al., 2025a) incorporated a sparse MoE architecture that routes different diffusion steps to specialized experts, while VEO3 (DeepMind, 2025) extended the paradigm by integrating audio, achieving state-of-the-art performance. The success of MovieGen (Polyak et al., 2024), Seaweed (Seawead et al., 2025), Goku (Chen et al., 2025b), and Waiver (Zhang et al., 2025d) further demonstrates the potential of video generation and its broad impact on practical applications. These developments underscore video generation as one of the most dynamic and competitive frontiers in generative AI community.

### B.2    AUTO-REGRESSIVE DIFFUSION MODEL

Auto-regressive generation dominates the text domain, while diffusion models have become the standard for visual generation. Recent research explores how to combine these paradigms to duplicate the long-term planning capacity of large language models in vision generation. A straightforward solution (Zhou et al., 2024) is to jointly train an autoregressive (AR) model for text and a diffusion model for vision, but this leaves the visual side reliant solely on diffusion without benefiting from AR modeling. Inspired by block diffusion (Arriola et al., 2025), several works (Li et al., 2024b; Hu et al., 2024; Deng et al., 2024; Ren et al., 2025; 2024) explore AR–diffusion hybrids: MAR (Li et al., 2024b) disentangles the two, letting AR predict conditions and diffusion reconstruct tokens; ACDiT (Hu et al., 2024) integrates them via block-wise diffusion with autoregression across blocks, while CausalDiffusion (Deng et al., 2024) extends this to token-level autoregression. Extending these ideas to video is natural since frames form temporal chunks. FAR (Gu et al., 2025) generates each frame autoregressively; MarDini (Liu et al., 2024a) employs an AR planner to provide frame-level conditions, with diffusion recovering pixels for tasks such as video interpolation, video extension, and image-to-video generation. Beyond this, MAGI (Teng et al., 2025) and Skyreel (Chen et al., 2025a) remove the dual-model design, training under the strategy of diffusion forcing (Chen et al., 2024a), where later frames are assigned higher noise levels, thereby enabling infinite autoregressive inter-chunk prediction and high-quality inner-chunk diffusion generation. More recently, self-forcing (Huang et al., 2025) highlights a gap between training (real data diffused with noise) and inference (model-generated conditions), and proposes rollout-based training to align the two, leading to more robust long-term prediction.

### B.3    EFFICIENT ATTENTION FOR MULTIMODAL GENERATION.

Diffusion Transformers (DiT) have emerged as the mainstream architecture for visual content generation. Representative models include PixArt-$\alpha$ (Chen et al., 2023b), Stable Diffusion 3 (SD3) (Esser et al., 2024), and Flux (Labs, 2024), the latter demonstrating the potential of scaling DiT to 12B

parameters for high-resolution image synthesis.. To address the computational challenges of vanilla attention ($O(n^2)$)), various methods have replaced it with linear-complexity mechanisms. For instance, DiG (Zhu et al., 2024) uses gated linear attention, PixArt-$\Sigma$ (Chen et al., 2024b) designs key-value token compression, while LinFusion (Liu et al., 2024b) involves Mamba-based structure and SANA (Xie et al., 2025a) employs ReLU linear attention approaches to reduce computational overhead. With the rise of video generation, the computational demands of standard quadratic attention have become a major bottleneck. To address the high computational cost of 3D video attention, many existing works employ factorized spatial and temporal attention to reduce complexity (Chen et al., 2023a; Ho et al., 2022; Wang et al., 2025c; Singer et al., 2022). Other methods reduce attention complexity by selectively skipping certain token interactions (Xi et al., 2025; Yang et al., 2025b; Li* et al., 2025; Zhang et al., 2025a;b;c). Simultaneously, other models, such as Mamba-based architectures (Wang et al., 2025b; Gao et al., 2024), have explored state-space models and linear-complexity designs for efficient video generation. However, these methods either retain some quadratic complexity due to global self-attention layers or are limited to local attention. In contrast, our model maintains a constant-memory KV cache with global attention mechanism, enabling the generation of high-quality, minute-length videos.

## C  MORE IMPLEMENTATION DETAILS

### C.1  PIPELINE CONFIGURATION

As detailed in Table 7, our SANA-Video-2B model supersedes the original SANA (Xie et al., 2025a) architecture, including the diffusion transformer and a small decoder-only text encoder, to best utilize the pre-trained text-to-image model's weights. However, we introduce several key modifications to support video generation. We increase the FFN dimension from 5600 to 6720 and the head dimension from 32 to 112 to accommodate 3D RoPE, and we add a temporal convolution in the Mix-FFN module to enhance motion performance. To effectively capture latent features from both images and videos, our approach uses different VAEs based on resolution. For 480P videos, we leverage a Wan2.1-VAE (Wang et al., 2025a) to prioritize reconstruction quality with a lower compression rate (F8T4C16). In contrast, for high-resolution 720P videos, we fine-tune the DCAE (Chen et al., 2024c) into a more aggressive deep compression autoencoder, DCAE-V (F32T4C32), to facilitate more efficient training and inference. For conditional feature extraction, we follow SANA by using a small decoder-only LLM for efficient text processing. For our training strategy, we also employ multi-aspect augmentation to enable arbitrary aspect ratio generation and facilitate image-video joint training, allowing the model to generate both images and videos from a single architecture. The AdamW optimizer (Loshchilov & Hutter, 2017) is utilized with a weight decay of 0.03 and a constant learning rate of 5e-5. We use Accelerate FSDP (acc, 2022) for efficient sharded data parallel training. Our final model is trained on 64 H100 GPUs for approximately 12 days.

### C.2  DETAILED TRAINING BREAKDOWN

In this subsection, we provide a detailed breakdown of the data scale, composition, and training steps for the three main stages of the SANA-Video training pipeline. Our sources of video data include public high quality datasets, Pexels and Artlist, as well as internal dataset. We design filtering and captioning pipeline (Appendix E), and the overall data scale after filtering is O(10M).

**Stage 1: VAE Adaptation on Text-to-Image (T2I).** This stage aims to adapt the VAE for robust representation learning before tackling the video domain. In this stage, we use approximately 10M internal images and train the model for 20K iterations with a batch size of 512.

**Stage 2: Continue Pre-training of T2V Model.** This stage aims at transferring learned image knowledge to video generation, establishing initial motion and short-term temporal coherence gradually from low resolution (192p) to high resolution (480p or 720p). In this stage, we use all the filtered video text pairs from our data pipeline (approximately 10M 5-second length video pairs). The LinearDiT model is trained for 200K iterations with a batch size of 128.

**Stage 3: Autoregressive Block Training (Long Video Fine-Tuning).** This stage aims at fine-tuning the LinearDiT to generate long videos in an autoregressive manner, enabling few-step and minute-length generation. In this stage (Sec. 3.4, we first use approximately 1M 5-second length short video samples to fine-tune the causal linear attention and causal MixFFN for 10K iterations.

---

**Algorithm 1** Block Linear Diffusion Inference with Linear KV Cache

---

**Require:** KV cache
**Require:** Denoise timesteps $\{t_1, \ldots, t_T\}$, noise scheduler $\Psi$
**Require:** Number of blocks $M$
**Require:** Block-wise diffusion model $G_\theta$ ($G_\theta^{\text{KV}}$ returns cumulative sum of state $\sum S$, cumulative sum of key
    $\sum \phi(K)^T$ and conv cache $f$)
1: Initialize model output $\mathbf{X}_\theta \leftarrow []$
2: Initialize KV cache $\mathbf{KV} \leftarrow [\text{None}, \text{None}, \text{None}]$
3: **for** $i = 1, \ldots, M$ **do**
4:    Initialize $x_{t_T}^i \sim \mathcal{N}(0, I)$
5:    **for** $j = T, \ldots, 1$ **do**
6:        Set $\hat{x}_0^i \leftarrow G_\theta(x_{t_j}^i; t_j, \mathbf{KV})$
7:        **if** $j = 1$ **then**
8:            $\mathbf{X}_\theta.\texttt{append}(\hat{x}_0^i)$
9:            Update Cache $\mathbf{KV} \leftarrow G_\theta^{\text{KV}}(\hat{x}_0^i; 0, \mathbf{KV})$
10:       **else**
11:           Sample $\epsilon \sim \mathcal{N}(0, I)$
12:           Set $x_{t_{j-1}}^i \leftarrow \Psi(\hat{x}_0^i, \epsilon, t_{j-1})$
13:       **end if**
14:   **end for**
15: **end for**
16: **return** $\mathbf{X}_\theta$

---

Subsequently, we leverage self-rollout long training with few-step distillation (DMD (Huang et al., 2025)) for 9K steps to further improve performance and efficiency. The long training uses 200K prompts from VidProM dataset (Wang & Yang, 2024).

Table 7: Architecture details of the proposed SANA-Video.

| Model | Width | Depth | FFN | #Heads | #Param (M) |
|---|---|---|---|---|---|
| SANA-Video-2B | 2240 | 20 | 6720 | 20 | 2056 |

### C.3 BLOCK LINEAR ATTENTION DURING INFERENCE

We follows Self-Forcing (Huang et al., 2025) for autoregressive inference, with the KV cache update based on our design (Alg. 1). Specifically, we first initialize KV cache as empty and start to denoise the first block. After it is fully denoised, the attention state $\sum_0^0 S$, cumulative sum of keys $\sum_0^0 \phi(K)^T$ and conv cache $f$ will be stored. For the remaining blocks (*e.g.*, $n$-th block), they will use the existing KV cache to denoise the latent until clean and then update the cumulative attention state $\sum_0^n S$ and cumulative sum of keys $\sum_0^n \phi(K)^T$. Also, conv cache $f$ will be replaced with the new cache. Such update leverages the global while keeping the memory constant and small, making the long video generation efficient and effective.

## D MORE RESULTS

Please refer to our **project page** (https://nvlabs.github.io/Sana/Video/), for the qualitative comparison and our generation results.

### D.1 VAE COMPARISON

In Sec. 3.5, we analyze the differences and performance of various video VAEs. To select the VAE that best suits our small diffusion model, we conducted a generalization experiment. We hypothesize that a VAE with better reconstruction ability under perturbation will be a better fit, as the diffusion model's output during inference may be slightly different from the clean latent distribution seen during VAE training. Specifically, we add Gaussian noise to the encoded latent before decoding

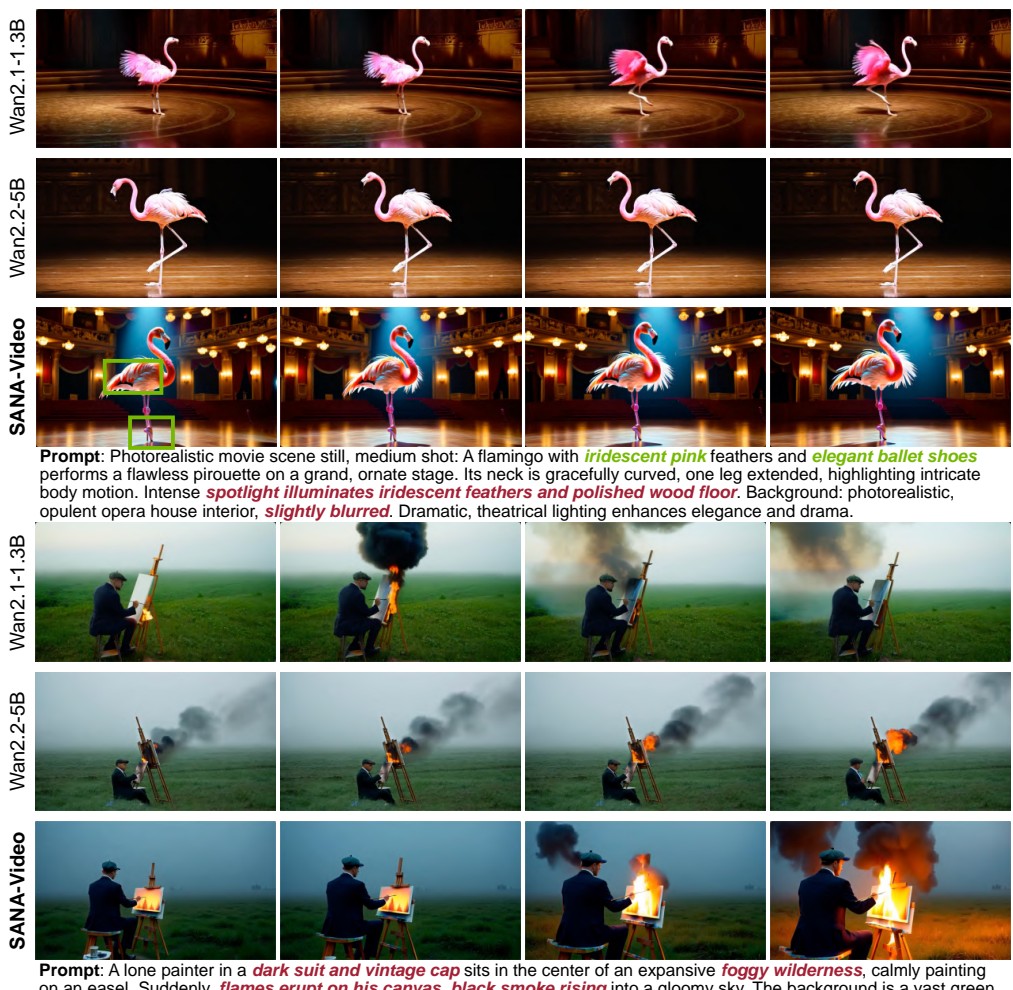

Figure 7: **Qualitative comparison among T2V methods.** SANA-Video has comparable motion control and video-text semantic alignment with state-of-the-art small diffusion models.

it, setting $x'_t = x_t + \epsilon z$, where $z \sim \mathcal{N}(0, I)$. As the results in Table 8 show, our DCAE-V performs much more robustly under different noise levels. This demonstrates its superior reconstruction generalization, making it the ideal choice for our small diffusion model.

Table 8: Performance comparison of different VAE models on 1000 samples from Panda-70M with different noise perturbation levels.

| Model | latent shape | psnr↑ | ssim↑ | lpips↓ |
|---|---|---|---|---|
| Wan2.1VAE ($\epsilon = 0$) | 16, T/4, H/8, W/8 | 34.41 | 0.95 | **0.01** |
| Wan2.2VAE ($\epsilon = 0$) | 48, T/4, H/16, W/16 | **35.61** | **0.96** | **0.01** |
| DCAE-V ($\epsilon = 0$) | 32, T/4, H/32, W/32 | 33.25 | 0.94 | 0.03 |
| Wan2.1VAE ($\epsilon = 0.1$) | 16, T/4, H/8, W/8 | 28.61 | 0.89 | 0.06 |
| Wan2.2VAE ($\epsilon = 0.1$) | 48, T/4, H/16, W/16 | 30.12 | 0.92 | **0.04** |
| DCAE-V ($\epsilon = 0.1$) | 32, T/4, H/32, W/32 | **31.91** | **0.93** | **0.04** |
| Wan2.1VAE ($\epsilon = 0.2$) | 16, T/4, H/8, W/8 | 24.25 | 0.78 | 0.16 |
| Wan2.2VAE ($\epsilon = 0.2$) | 48, T/4, H/16, W/16 | 25.94 | 0.84 | 0.10 |
| DCAE-V ($\epsilon = 0.2$) | 32, T/4, H/32, W/32 | **29.34** | **0.90** | **0.05** |

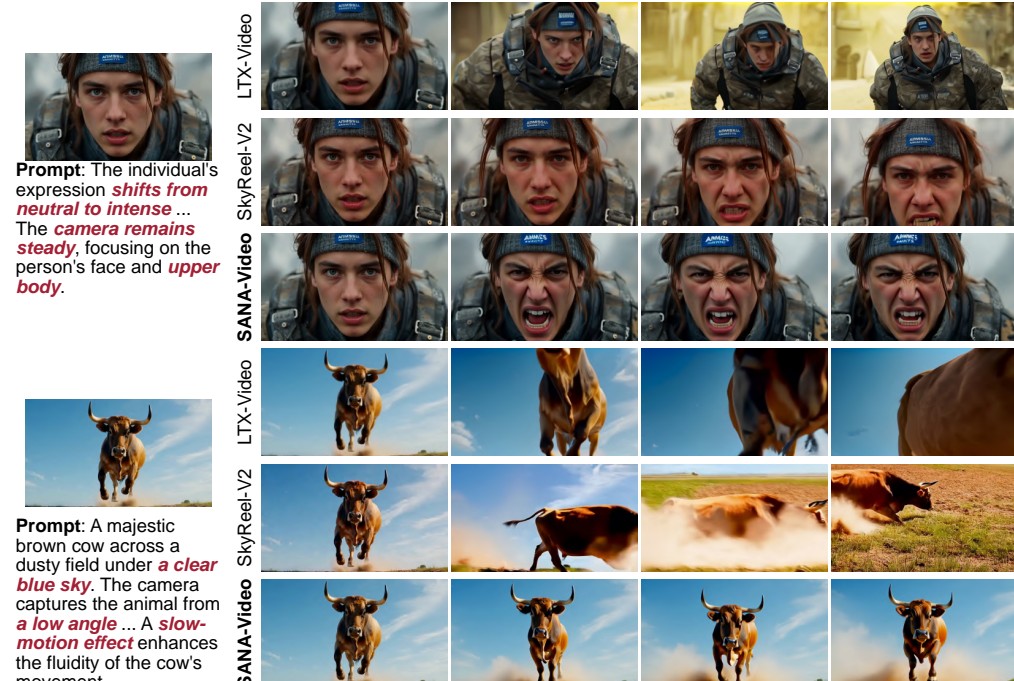

Figure 8: **Qualitative comparison among I2V methods.** SANA-Video has better motion control and video-text semantic alignment.

## D.2    QUALITATIVE COMPARISON

**Text-to-Video Generation.** We compare the text-to-video generation results with current state-of-the-art small diffusion models Wan2.1-1.3B (Wang et al., 2025a) and Wan2.2-5B (Wang et al., 2025a). As shown in Fig. 7, SANA-Video has comparable semantic understanding, great motion control, and high aesthetic quality.

**Image-to-Video Generation.** We compare the image-to-video generation results with small diffusion models LTX-Video (HaCohen et al., 2024) (2B) and SkyReelv2-I2V (Chen et al., 2025a) (1.3B). As shown in Fig. 8, SANA-Video has the best semantic understanding ability ("camera remains steady" instruction in the first case) as well as the best motion control ("slow-motion effect" instruction in the second case) and moderate motion magnitude (first case).

## D.3    MORE I2V RESULTS

Our SANA-Video is a unified framework that can perform T2I, T2V and I2V with a single model. We visualize the I2V generation results in Fig. 9. The first column is the reference image and the remaining columns are the generated video. Our SANA-Video can generate semantic consistent and temporal smooth videos based on the first frame.

## D.4    INFLUENCE OF MOTION SCORE

As mentioned in our data pipeline (Sec. E), we use the average optical flow value to represent the motion magitude, which is called motion score in our paper. The motion score is added to the text prompt to better control the motion. In Fig. 10, we compare the impact of motion score in the I2V task, which is more clear with the same reference image. By increasing the motion, SANA-Video can generate videos with larger but still consistent motion.

**Reference Frame**                **Generation Frames**

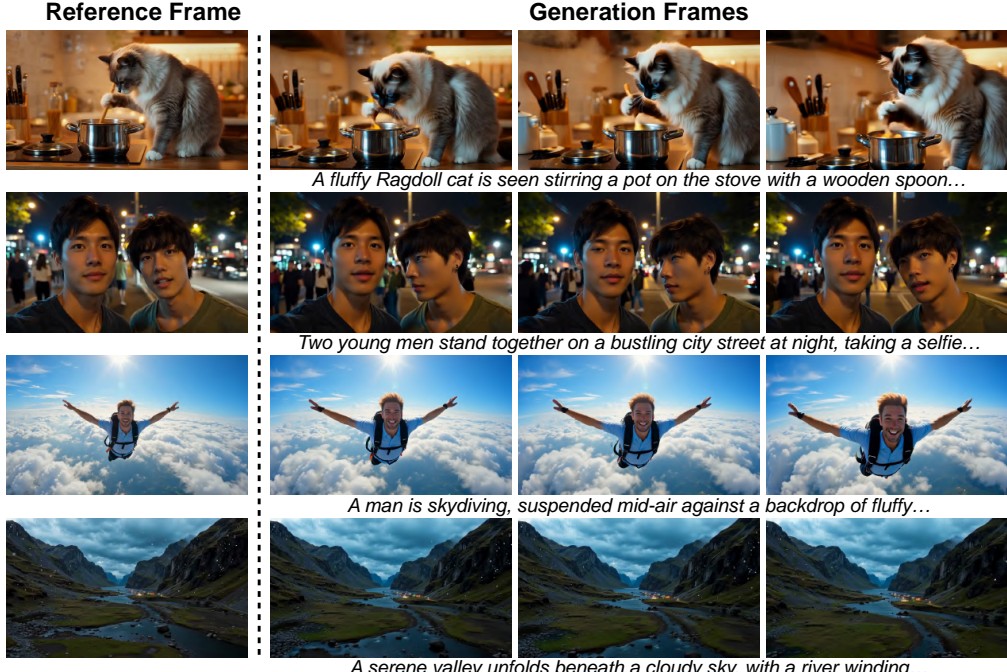

Figure 9: **Visualization of image-to-video generation.** SANA-Video can keep consistent with the first frame while generating realistic motion.

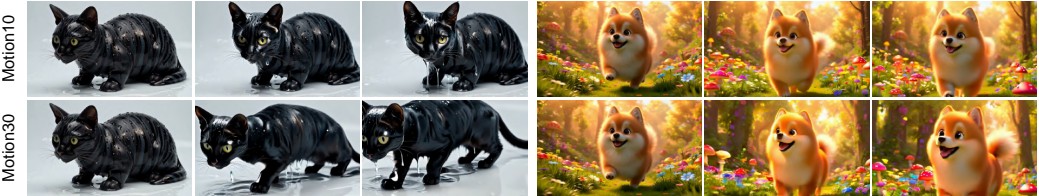

Figure 10: **The impact of motion score on I2V task.** Higher motion score can lead to larger motion.

### D.5 LONGSANA VISUALIZATION

In Fig. 11, we provide an example of our 1-minute long video generation. LongSANA is able to generate motion consistent and semantically aligned long videos.

### D.6 LONG VIDEO EVALUATION

We further conduct quantitative evaluation on VBench-Long (Zhang et al., 2024) to verify the effectiveness of LongSANA in long video generation. We compare LongSANA with previous state-of-the-art methods on 30-second video generation in Table 9. Our LongSANA achieves the best semantic and total scores. In addition, SANA-Video is the fastest method that can generate videos in real-time with 27.5 FPS, demonstrating the efficiency and effectiveness of LongSANA when handling long video sequences.

## E DATA PROCESSING PIPELINE

To build our training dataset, we collect public real and synthetic data and implement a multi-stage filtering paradigm. First, we use PySceneDetect (Castellano) and FFmpeg (Developers, 2025) to cut raw videos into single-scene short clips. For each video clip, we analyze its aesthetic and motion quality, as well as providing detailed captions. Specifically, the motion quality is measured by Unimatch (Xu et al., 2023) (optical flow) and VMAF (Peng et al., 2025) (pixel difference), and

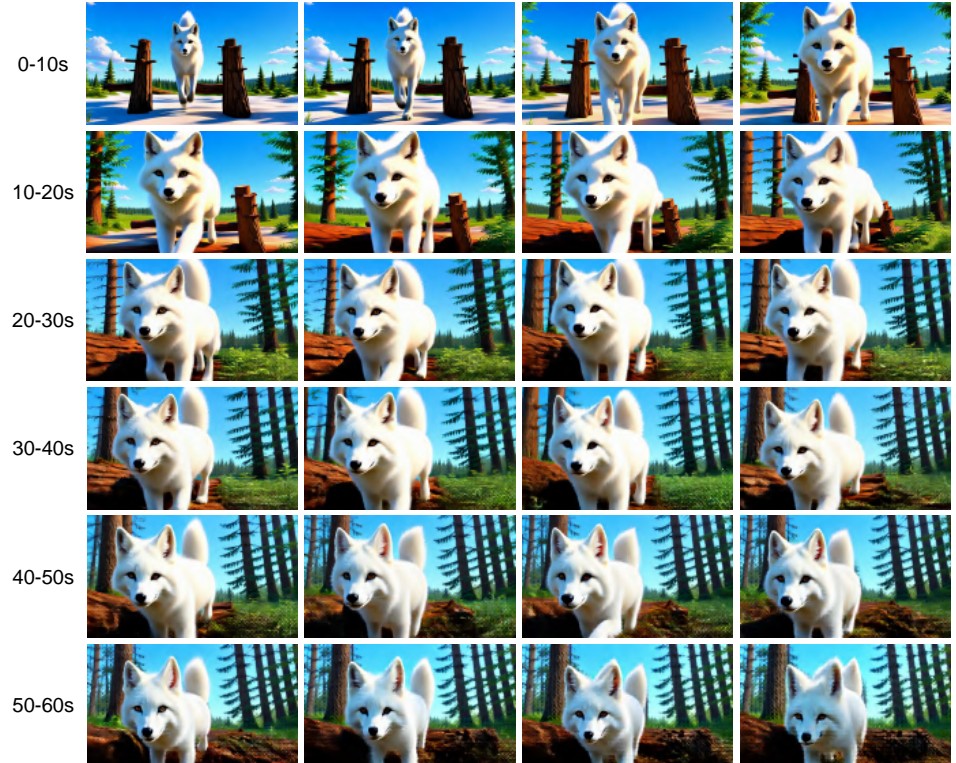

| | 0-10s | | | |
| 10-20s | | | |
| 20-30s | | | |
| 30-40s | | | |
| 40-50s | | | |
| 50-60s | | | |

*A white Arctic fox runs gracefully across a fallen log in a dense forest. The fox's fur is pristine white, and it moves with agility and purpose, its tail held high. The camera follows the fox closely, capturing its fluid movements and the serene beauty of the woodland setting. The fox's ears are perked up, and its eyes are focused ahead, embodying the spirit of freedom and wildness.*

Figure 11: **Long video visualization of LongSANA.**

Table 9: **Comparison of long video generation methods on the VBench-Long (Zhang et al., 2024) benchmark.** All compared methods generate 30s videos for evaluation.

| Model | Total Score ↑ | Quality Score ↑ | Semantic Score ↑ | FPS |
|---|---|---|---|---|
| SkyReels-V2 (Chen et al., 2025a) | 75.29 | 80.77 | 53.37 | 0.49 |
| FramePack (Zhang & Agrawala, 2025) | 81.95 | 83.61 | 75.32 | 0.92 |
| Self Forcing (Huang et al., 2025) | 81.59 | 83.82 | 72.70 | 17.00 |
| **SANA-Video** | **82.29** | 83.10 | **79.04** | 27.50 |

only clips with moderate and clear motion are kept. Furthermore, the average optical flow is used as a representation of motion magnitude, injecting into prompt for better motion controllability. Aesthetic quality is measured by a pre-trained video aesthetic model (DOVER (Wu et al., 2023a)) and key frame saturation obtained with OpenCV (Bradski, 2000), where low aesthetic score and over-saturated videos are removed. Finally, we collect approximately 5,000 human preferred high-quality videos based on stringent motion and aesthetic criteria. The SFT data is collected with diverse but balanced motion and style categories, which can further improve the overall performance. As shown in Fig. 12, the details of this data processing pipeline are discussed as follows.

**Scene Detection and Shot Cut.** In the pre-training stage, we focus on generating 5-second short videos with 16 FPS on a specific scene. However, the raw videos are commonly long and contains more than one scene. Therefore, we cut the raw videos to small video shots with two steps: PySceneDet (Castellano) to split the scenes and FFmpeg (Developers, 2025) to split videos into short clips.

**Motion Filtering.** Our pre-training dataset comes from multiple sources, and each source of data differs not only in style but also in motion. Motion that is too fast or too slow degrades the motion performance of SANA-Video. Following Zheng et al. (2024), we apply Unimatch (Xu et al., 2023)

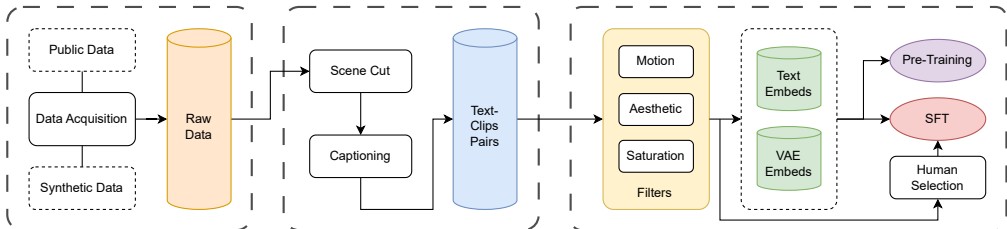

Figure 12: Data filtering paradigm of SANA-Video.

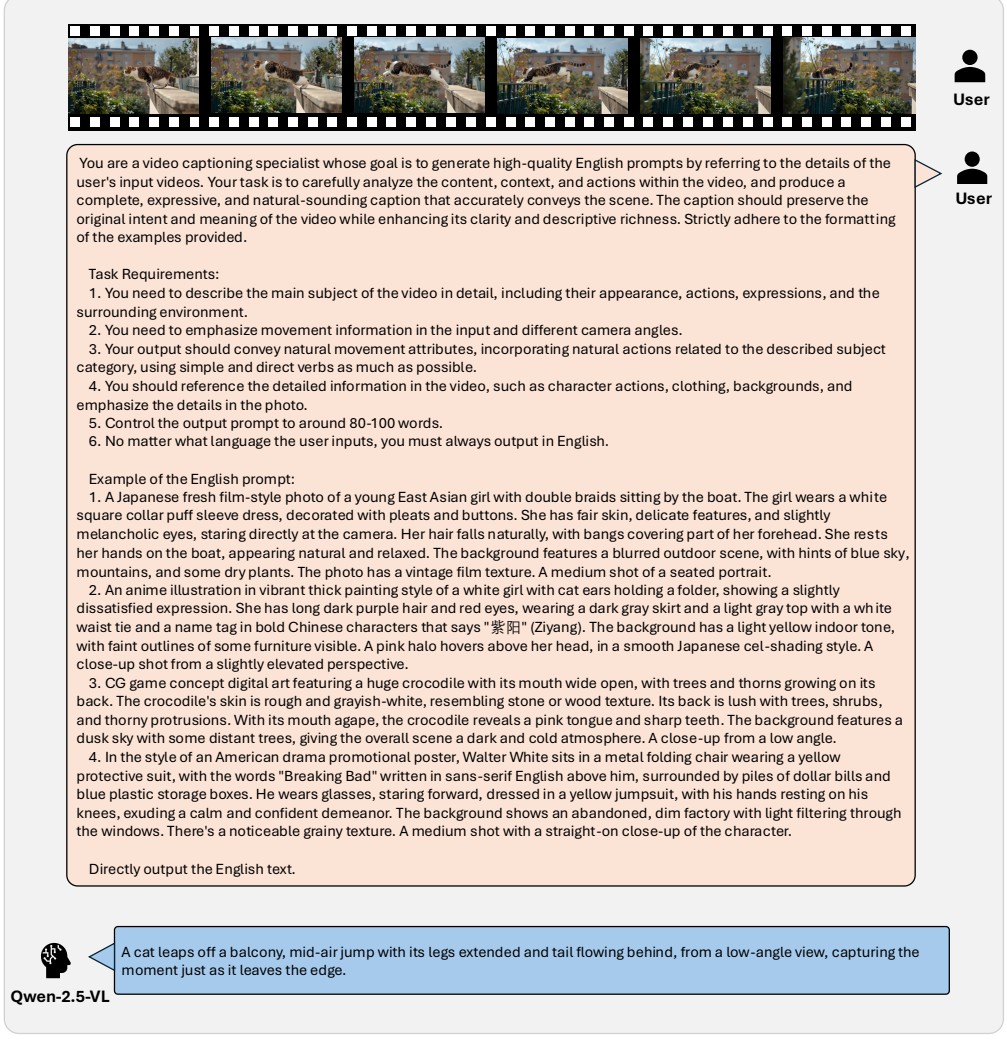

Figure 13: An overview of the captioning pipeline.

and Vmaf to score the motion of each video. Unimatch can evaluate the optical flow of two given images of the same shape. We select frames from each video every 0.5 seconds, reshape them into 320x576, and calculate the average optical flow over all selected frames. Vmaf, on the other hand, simply computes the pixel difference of two images; we use FFmpeg (Developers, 2025) to compute the Vmaf over all consecutive frames and normalize them. Due to the variance of different video sources, we analyze the motion scale and set the appropriate motion range individually, ensuring our data has moderate and clear motion. During pre-training, we also append *Motion score: {unimatch value}* to the text prompt to help control the motion magnitude of the generated videos (Fig. 10).

**Aesthetics** Many Text-to-Image works have proven that high aesthetic data can improve the training efficiency of an image generation model (Chen et al., 2023b). We believe that this also applies to

the video generation model. We use Dover (Wu et al., 2023a) to score each video for its aesthetics. Dover produces three different scores: aesthetic score, technical score, and overall score, among which we use the overall score as the filter metric.

**Saturation** We also observe that some of our data, especially synthetic data and real data converted from HDR to SDR, has unnatural color, appearing in high saturation. To prevent these data from damaging the output quality of SANA-Video, we use OpenCV (Bradski, 2000) to compute a saturation score of each video. We select frames from each video every 0.5 seconds, convert their color representation from RGB to HSV, where the "S" channel in HSV color representation stands for saturation. By averaging the "S" channel over all pixels and frames, we obtain the saturation score of a video. We keep only videos with a saturation score lower than a threshold set to a reasonable value for each data source.

**Captioning** (Wang et al., 2025a) shows that LLM rewritten prompts can produce more accurate and detailed prompts within the same distribution, and thus make the model easier to learn, and consequently enhance the model's performance. Moreover, for synthetic data with existing prompts, replacing their prompts with LLM rewritten ones helps reduce the misalignment between the original prompts and their synthetic output. Following (Wang et al., 2025a), we use Qwen-2.5-VL (Bai et al., 2025b) to caption our data as shown in Fig. 13.

**SFT Data** For our final stage of SFT training, we selected approximately 5,000 high-quality videos based on stringent criteria for motion and aesthetics. The motion requirement is fulfilled by the presence of either distinct object motion, camera motion, or both. Ideal object motion is characterized by a moderate magnitude and a clearly focused action that is free from occlusions. Similarly, any camera movement must be stable and smooth, without jittering, to maintain 3D consistency. The aesthetic criteria are equally comprehensive. Beyond technical qualities like balanced brightness and natural color, we prioritize videos with appealing overall content and layout, demonstrated by thoughtful composition and engaging subject matter. Following this filtering process, the videos were classified into four motion categories (human activities, animal activities, other objects, natural or urban scenes) and three aesthetic styles (realistic, cartoon, cinematic). This strategic sampling across diverse categories is crucial for ensuring both the model's high performance and the breadth of its capabilities. The influence of fine-tuning on the SFT data is illustrated in Fig. 14, where both the aesthetic details (the eyes in the first example), and the motion realism (the pipe of the second example) will be improved.

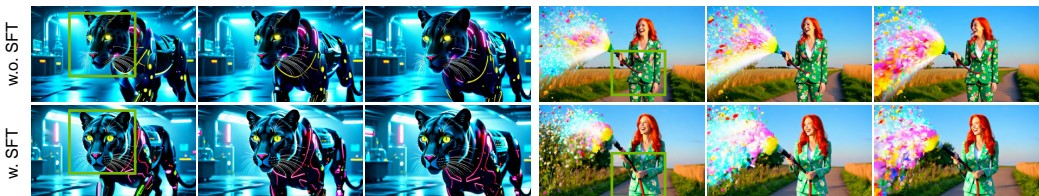

Figure 14: **Analysis of the influence of SFT.** Fine-tuning on the human preferred SFT data can improve the video details and adherence to the laws of physics.

## F  WORLD MODEL

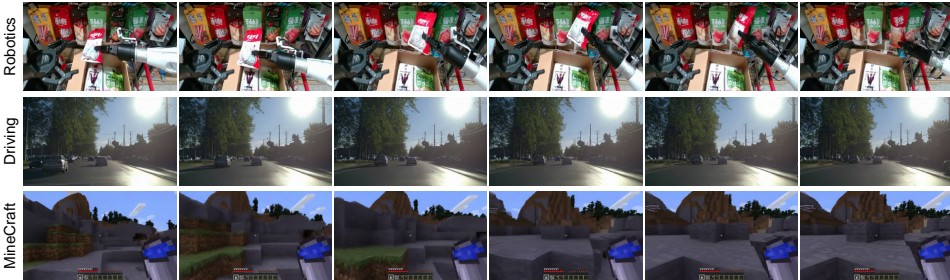

Figure 15: **Visualization of world model task generation.**

We fine-tune SANA-Video on several downstream tasks to demonstrate the potential of applying SANA-Video to world model related generation: embodied AI, autonomous driving and game generation.

**World Model for Embodied AI.** The first important downstream task for video generation is embodied AI, where SANA-Video can be used to generate simulation data for robot training. In this task, we leverage AgiBot (contributors, 2024) as the training data, which contains synchronized views of different camera views. The head-front view is adopted as the target videos and filtered with our data pipeline. The generation results are shown in the first row of Fig. 15.

**World Model for Autonomous Driving.** Video generation model is also a good simulator for autonomous vehicle scenarios, and SANA-Video can be used to generate diverse and realistic driving scenes. In this task, we fine-tune SANA-Video on internal driving data, using the front camera with 30 FOV. The generation results are shown in the second row of Fig. 15.

**World Model for Game Generation.** We explore downstream game generation to create interactive video games. Specifically, we use VPT (Baker et al., 2022) as the training data, containing screen recording videos of players playing Minecraft. The raw videos are cut and processed following our data pipeline in Appendix E. In addition, we train a small classifier to identify low-quality data in the scenario. The generation results are shown in the third row of Fig. 15.

