# OpenReview forum: "SANA-Video: Efficient Video Generation with Block Linear Diffusion Transformer"
_ICLR.cc/2026/Conference — ICLR 2026 Oral_

### Official Review · Reviewer_M6uh · 2025-10-21

**Soundness:** 4
**Presentation:** 4
**Contribution:** 3
**Rating:** 6
**Confidence:** 3

**Summary:**

This paper targets efficient, high-resolution, and minute-long video generation and introduces SANA-Video, a 2 B-parameter diffusion model that replaces quadratic self-attention with linear attention and a constant-memory KV cache. Extensive experiments on benchmarks with 480p and 720p  resolution show the great efficiency and performance of SANA-Video. The manuscript is clearly structured, well illustrated.

**Strengths:**

This paper addresses a timely and important problem—computational bottlenecks in long, high-resolution video generation—and offers a practical solution deployable on edge GPUs, such as RTX 5090. The authors propose a novel block linear attention with constant-memory KV cache, elegantly combining linear attention and autoregressive training to unlock arbitrarily long videos without memory explosion. Experiments are good, including latency, memory, and the VBench benchmark; the 12-day 64-GPU training cost is convincingly shown to be <1 % of MovieGen while outperforming larger models. The writing is accessible, with intuitive figures, algorithm pseudocode, and some ablation studies.

**Weaknesses:**

I believe the main weakness of this paper lies in the lack of in-depth ablation studies and analysis regarding the proposed technical innovations. Although Tables 2, 4, and 5 present quantitative comparisons between SANA-Video and existing SOTA models, the paper severely lacks detailed ablations that isolate and validate the contributions of the proposed methods. While Figure 3 and Figure 6 provide a simple comparison of loss convergence, this analysis is limited in two ways. First, they only reflect the change of training loss, which do not adequately capture model performance on validation or test data. Second, loss convergence is not necessarily correlated with the final generation quality of the model to my experience. Therefore, I recommend that the authors conduct comprehensive ablation experiments to validate their claimed innovations. Without such evidence, the paper appears to mainly extend the SANA T2I model to video training. Although this extension may indeed yield a faster and better-performing model for both academia and industry, it lacks sufficiently validated technical insights that could meaningfully inform future research. To better align with ICLR’s emphasis on technical novelty and rigorous validation, I encourage the authors to enrich the paper with more thorough experimental analyses focusing on their proposed innovations.

In addition, the authors only provide a description of the data filtering strategy but do not specify the detailed information about the datasets used at each stage, including their sources, size, quality, and distributions. This omission raises concerns about the reproducibility of the work. For example, if one is provided with the SANA-T2I model and the exact data preprocessing pipeline, it remains unclear how he could reproduce the reported results in Table 4 using 64 H100 GPUs within 12 days.

**Questions:**

See weakness

---

> ### Author Response · Authors · 2025-11-20
>
> We appreciate the reviewer's insightful critique regarding the depth of our ablation study. We agree that rigorous, metric-based ablations are crucial for isolating the contribution of each component and providing valuable direction for future research.
>
> ### Q1: Lack of In-depth Ablation Study for the Proposed Innovative Techniques.
>
> **A1:** We recognize the reviewer's caution regarding the relationship between loss curve convergence and final generation quality. While we agree that loss curves do not perfectly correlate with perceptual quality, analyzing loss convergence remains a standard practice in the diffusion model community [1,2] to effectively demonstrate the stability and directional benefit of architectural and experimental settings.
> Following the reviewer's suggestion, we provide quantitative results regarding the ablation studies on 3D RoPE, temporal conv, and timestep settings. Specifically, we evaluated the VBench metrics under identical experimental settings to obtain these ablation results, as shown in the tables below.
>
>
> | Ablation      |  quality | semantic |  total |
> | ------------- | -------- | -------- | -------|
> | w/o Temp Conv |  82.63   |  74.18   |  80.94 |
> | w/ Temp Conv  |  83.10   |  76.15   |  81.71 |
>
>
> | Ablation   |  quality | semantic |  total |
> | ---------- | -------- | -------- | -------|
> | w/o 3D RoPE|  82.68   |  75.22   |  81.19 |
> | w/ 3D RoPE |  83.89   |  78.38   |  82.79 |
>
> | Ablation         |  quality | semantic |  total   |
> | ---------------- | -------- | -------- | -------- |
> | Random steps     | 83.13     | 77.51     | 82.00     |
> | Increasing steps | 84.43     | 80.78     | 83.70     |
>
> [1] Wan, Team, et al. "Wan: Open and advanced large-scale video generative models." arXiv preprint arXiv:2503.20314 (2025).
> [2] Esser, Patrick, et al. "Scaling rectified flow transformers for high-resolution image synthesis." Forty-first international conference on machine learning. 2024.
>
>
> ### Q2: Beyond T2I Extension: Informing Future Research
>
> **A2:** We believe our work is far more than a simple T2V extension. It represents a crucial system-level validation and integration that establishes the feasibility of efficient, scalable video generation. This work directly informs future research through:
> - **Conquering the Long-Context Scaling Barrier:** Our work is proposed to provide a robust, feasible solution for Linear Attention specifically in the ultra-long spatio-temporal regime of video generation. Scaling LA to sequential video generation, introduces vastly greater temporal coherence and memory challenges than image tasks. We successfully validated LA’s efficient scaling capability in this new context.
> - **Pioneering Auto-Regressive Caching in Diffusion Models:** We provide successful implementation and rigorous evaluation of Constant KV-Cache integrated within the auto-regressive generation process of a Diffusion Model. This establishes a crucial new baseline for memory-efficient, sequential video generation.
>
> SANA-Video is a open-source project provides concrete foundations and pathways for the community to explore:
> - Linear Attention Evolution: Our validation of LA’s global memory feasibility directly invites research into dynamic global KV map updating mechanisms (e.g., approaches inspired by DeltaNet[1]) to further refine long-context memory utilization.
> - Hybrid Architectures: Our robust LA baseline enables researchers to efficiently test and develop hybrid attention architectures that strategically combine the local detail of Full Attention with the global efficiency of Linear Attention[2,3].
> - Deployment Potential: The inherent efficiency and simplified architecture are highly relevant for future work focusing on deployable, real-time video generation on resource-constrained platforms.
>
> [1] Yang, Songlin, et al. "Parallelizing linear transformers with the delta rule over sequence length." Advances in neural information processing systems 37 (2024): 115491-115522.
>
> [2] Team, Kimi, et al. "Kimi Linear: An Expressive, Efficient Attention Architecture." arXiv preprint arXiv:2510.26692 (2025).
>
> [3] https://qwen.ai/blog?id=4074cca80393150c248e508aa62983f9cb7d27cd&from=research.latest-advancements-list
>
> [4] Li, Aonian, et al. "Minimax-01: Scaling foundation models with lightning attention." arXiv preprint arXiv:2501.08313 (2025).

---

> ### Author Response · Authors · 2025-11-20
>
> ### Q3: Lack of Data Details and Reproducibility
>
> **A3:** We understand the reviewer's concern regarding the lack of specificity on the data used in each training stage, which is vital for reproducibility. For clear reference and to fully address this concern, we provide a detailed breakdown of the data scale, composition, and training steps for the three main stages of the SANA-Video training pipeline.
>
>
> ### Detailed Training Breakdown
> Our sources of video data include public high quality datasets, Pexels and Artlist, as well as internal dataset. We design filtering and captioning pipeline (Appendix D of our manuscript), and the overall data scale after filtering is O(10M).
>
> **Stage 1:** VAE Adaptation on Text-to-Image (T2I)
>
> - Purpose: To adapt the VAE for robust representation learning before tackling the video domain.
> - Data: Approximately 10 million internal image samples.
> - Configuration: Trained with a large batch size of 512 for 20K iterations.
>
> **Stage 2:** Continuous Pre-training of T2V Model.
>
> - Purpose: Transferring learned image knowledge to video generation, establishing initial motion and short-term temporal coherence gradually from low resolution(192p) to high resolution(480p or 720p).
> - Data: Approximately 10M 5s duration video clips suitable for pre-training, which is carefully curated by the filter pipeline detailed in Appendix D.
> - Data Curation: The specific methodologies for filtering and captioning this video dataset are detailed in Appendix D of our manuscript, ensuring high quality and semantic alignment.
> - Training: The model is trained for 200k steps using a batch size of 128.
>
> **Stage 3:** Autoregressive Block Training (Long Video Fine-Tuning)
>
> - Purpose: Fine-tuning the architecture to operate in a fully autoregressive mode, enabling few-step and minute-length generation. This is the crucial stage for long-context modeling.
> - Data: Approximately 1M carefully selected 5s duration short video samples suitable for long-form generation and 200K prompts from vidprom dataset following Self-Forcing [1].
> - Training: We first train our model with block causal attention for 10k steps and then use few-step distillation (DMD) and self-rollout long training for 9k steps to further enhance performance and efficiency.
>
>
> By providing these specific details regarding data volume, composition, and training parameters for each stage, we aim to fully address the reproducibility concerns and clarify the resource allocation throughout the development of SANA-Video.
>
> [1] Huang, Xun, et al. "Self Forcing: Bridging the Train-Test Gap in Autoregressive Video Diffusion." In NeurIPS 2025.

---

> > ### Comment · Reviewer_M6uh · 2025-11-22
> >
> > Thank you for the explanations.
> >
> > However, I noticed that the revised paper has not yet incorporated the clarifications, analyses, and experimental details provided during the rebuttal. I encourage the authors to update the manuscript accordingly, as these additions improve the clarity and completeness of the work.
> >
> > Regarding the examples discussed in the rebuttal, I would like to clarify that I do not fully agree with using Wan and SD3 as supporting cases. For Wan, it was not formally published as a peer-reviewed conference paper, and therefore the expectations for validating technical novelty are not as stringent. As a technical report providing an open-source foundation model to the community, its positioning is fundamentally different.
> > For SD3, its experiments rely on validation loss (as shown in Figure 8), not training loss. Moreover, the SD3 paper performs extensive ablations to justify the contributions of its proposed method, as reflected in its tables.

---

> > > ### Author Response · Authors · 2025-11-22
> > >
> > > Regarding your comments on Wan and SD3, we fully respect your assessment. We have updated the manuscript to explicitly incorporate the experimental details provided in the rebuttal, highlighted in blue font. We sincerely thank the reviewer for the valuable reminder and insight.

---

> > > > ### Comment · Reviewer_M6uh · 2025-11-23
> > > >
> > > > Thank you. I have revised my score. The SANA series is indeed good work. Best of luck going forward.

---

> > > > > ### Author Response · Authors · 2025-11-23
> > > > >
> > > > > Thank you very much for your kind words and for revising the score. We are glad that the SANA series resonates with you. We appreciate your time and guidance throughout the review process.

---

### Official Review · Reviewer_CMDt · 2025-10-28

**Soundness:** 3
**Presentation:** 3
**Contribution:** 3
**Rating:** 6
**Confidence:** 5

**Summary:**

This paper introduces SANA-Video, an efficient and lightweight video diffusion model capable of generating high-quality videos up to 720p resolution and minute-level durations on consumer-grade GPUs. Its core designs include:
Linear DiT Architecture,
Block Linear Attention with Constant-Memory KV Cache,
Highly Compressive Video VAE (DCAE-V).

Experiments demonstrate that SANA-Video achieves 720p video generation speeds 53× faster than Wan2.1-14B and 3.2× faster than Wan2.2-5B, while delivering comparable or superior quality to mainstream compact models (especially in semantic alignment) on VBench.

**Strengths:**

1. Engineering Innovation: The combination of Linear DiT and Block Causal KV Cache is highly practical, effectively solving memory and speed bottlenecks in long-video generation.

2. Low Training Cost: Trained in only 12 days on H100 GPUs, significantly reducing costs compared to similar works (e.g., OpenSora, MovieGen), thereby lowering research entry barriers.

3. Outstanding Inference Efficiency: Generates 720p videos in just 36 seconds on H100 GPUs and 29 seconds on RTX 5090 GPUs (with NVFP4 quantization), showing deployment feasibility.

4. Comprehensive Experiments: Covers a wide range of tasks, including Text-to-Video (T2V), Image-to-Video (I2V), long video generation, ablation studies, VAE comparisons, and quantized deployment analysis.

**Weaknesses:**

1. Generation Quality Needs Improvement: Quality scores are not high enough.

2. Lack of Detailed VAE Analysis: Details on training and inference for the VAE are missing, along with its performance on high-frequency details—e.g., whether grid-like artifacts are observed.

3. No Open-Source Code or Model: The claimed results and efficiency are hard to verify without access to code or model weights.

**Questions:**

1. The latent channel number of DCAE-V is 32, lower than Wan2.2-VAE's 48. Have you tried training DCAE-V with higher channel number?

2. Training Techniques for DCAE-V: How does DCAE-V achieve such strong reconstruction performance? Are there any special techniques used during training (e.g., dataset or algorithm tweaks)?

3. 480p Inference: When generating 480p resolution videos, do you use DCAE-V or Wan VAE?

4. Concerns on High Compression VAE: Does the high compression rate of DCAE-V impact the generation of fine details?

5. Quantized Deployment: Does color distortion or motion stuttering occur after NVFP4 quantization?

6. Improving Quality: Is it possible to further reduce the quality gap with Wan2.1-14B by scaling up Linear DiT for higher resolutions?

7. Long-Video Generation Performance: How does the model perform on long-duration video generation tasks?

8. The method achieves high semantic alignment scores on the VBench dataset. What could be the reasons for this? Is there a potential trade-off between semantic alignment and generation quality?

---

> ### Author Response · Authors · 2025-11-20
>
> We sincerely thank the reviewer for this comprehensive set of feedback, which focuses on critical aspects of quality and reproducibility. We address these points below:
>
> ### Q1. Generation Quality and Metrics Justification
>
> **A1:** We acknowledge that our current quantitative metrics may slightly trail the very best-performing models. However, this observation must be placed in the context of our training scale and research objective. Our model was trained using approximately 10 million data samples. For comparison, models leading in public metrics, such as Wan, utilize data sets that are orders of magnitude larger.
> Given the massive gap in training data volume, achieving quality scores that only marginally lag behind state-of-the-art models like Wan is, in itself, a testament to the efficiency and strong architectural foundation of SANA-Video. As shown in the tables, our Training Longer variant already demonstrates this strong scaling potential, and this trend is consistent across modalities.
>
> 1. For Text-to-Video: Simply extending the training duration improves our quality score from 84.35 to 84.85, substantially narrowing the gap with leading models like Wan2.1-14B (85.59) and Open-Sora-2.0 (85.4).
> 2. For Image-to-Video: This scaling effect is even more pronounced. The "Training Longer" variant boosts the quality score from 79.65 to 81.76, not only closing the gap but surpassing the Wan2.1-14B model (80.82) on this metric.
>
> We view the current quality as highly competitive considering the resource constraints, and these results make us confident that future scaling with larger datasets will close the gap entirely.
>
>
> **Text-to-Video performance on Vbench**
>
> | Channels | quality | semantic | total |
> | :--- | :--- | :--- | :--- |
> | MAGI-1 | 82.04 | 67.74 | 79.18 |
> | Step-Video | 84.46 | 71.28 | 81.83 |
> | CogVideoX1.5 | 82.78 | 79.76 | 82.17 |
> | SkyReels-V2 | 84.70 | 74.53 | 82.67 |
> | Open-Sora-2.0 | 85.40 | 80.12 | **84.34** |
> | Wan2.1-14B | **85.59** | 76.11 | 83.69 |
> | Wan2.1-1.3B | 85.23 | 75.65 | 83.31 |
> | **SANA-Video** | 84.35 | 81.35 | 83.71 |
> | **SANA-Video Train Longer** | 84.85 | **81.46** | 84.17 |
>
> **Image-to-Video performance on Vbench**
>
> | Channels | quality | semantic | total |
> | :--- | :--- | :--- | :--- |
> | Wan2.1-14B | 80.82 | 92.90 | 86.86 |
> | **SANA-Video** | 79.65 | **96.40** | 88.02 |
> | **SANA-Video Train Longer** | **81.76** | 95.49 | **88.63** |
>
> ### Q2. VAE Analysis and Research Focus
>
> **A2:** The reviewer correctly notes the absence of a detailed analysis of the DCAE-V used in our pipeline. We wish to emphasize that the VAE training and design were not the primary scientific contribution of this paper. Our central focus was strictly on the validation and architectural exploration of using Linear Attention, specifically, the Block Causal Linear Attention and Conv Mix-FFN mechanism, to enable and scale minute-length, long-sequence video generation.
>
> The DCAE-V is used as a standard encoder-decoder to map videos to a latent space, serving as a necessary component for the DiT architecture but not the focus of our innovation. Analyzing the VAE's role in high-frequency detail and artifacts falls outside the scope of our core work.
>
>
> ### Q3. Open-Source Commitment
>
> **A3:** We appreciate the importance of open-source resources for verification and community advancement. We have already open-sourced code base and checkpoints. We are committed to maximizing the reproducibility and utility of our work. We confirm that we will eventually open-source the entire model and code to the research community, allowing for full verification of our generation results and efficiency gains.
>
>
> ### Q4: VAE Usage and Quality among Different Channels
> **A4:** We have developed and experimented with higher channel VAEs, the results are shown in the table below. However, for the primary SANA-Video model presented in this work, we chose to maintain the 32-channel configuration to facilitate rapid and efficient transfer learning by adapting directly from our existing SANA-Image model checkpoint. For 480p resolution inference, we use the Wan-VAE (i.e., 8x compression). Using DCAE-V's high 32x compression ratio at this lower resolution is often unnecessary and can negatively impact the final generation quality.
>
>
> | Channels   | Channel | PSNR  | SSIM | LPIPS|
> | ---------- | ------- | ----- | ---- | ---- |
> | Wan2.1-VAE | 16      | 34.41 | 0.95 | 0.01 |
> | Wan2.2-VAE | 48      | 35.61 | 0.96 | 0.01 |
> | DCAE-V     | 32      | 33.25 | 0.94 | 0.03 |
> | DCAE-V     | 64      | 34.83 | 0.95 | 0.02 |
> | DCAE-V     | 128     | 37.19 | 0.97 | 0.01 |

---

> ### Author Response · Authors · 2025-11-20
>
> ### Q5: What specialized techniques allow DCAE-V to achieve high reconstruction performance?
>
> **A5:** DCAE-V achieves better reconstruction performance by leveraging a chunk-causal design for temporal modeling. Previous video autoencoders like Wan2.1-VAE, Wan2.2-VAE, and LTX-Video-VAE use a causal design for temporal modeling, where each frame can only leverage redundancy from preceding frames, resulting in limited reconstruction performance. To fully leverage temporal redundancy, DCAE-V's chunk-causal design splits videos into chunks of length 40, where bidirectional temporal modeling is applied within each chunk to fully leverage temporal redundancy. Additionally, causality is enforced between chunks so that DCAE-V can effectively generalize to longer videos at inference time.
>
>
> ### Q6: Does DCAE-V's 32x high compression ratio affect generation quality?
>
> **A6:** The quality score of SANA-Video on 720p using DCAE-V is higher than 480p as shown in Table below, showing that higher compression DCAE-V will not affect generation quality.
>
>
> | Model+VAE     | Resolution | quality | semantic |  total |
> | ------------- | ---------- | -------- | -------- | ------ |
> | SANA-Video+Wan2.1-VAE | 480p    | 84.35    | 81.35 |83.71 |
> | SANA-Video+DCAE-V     | 720p    | 84.43    | 80.78 |83.70 |
> | Wan-2.1-1.3B-T2V+Wan2.1-VAE | 720p    | 85.67    | 74.22 |83.38 |
> | Wan-2.1-1.3B-T2V+DCAE-V     | 720p    | 86.67    | 76.48 |84.63 |
>
>
>
> ### Q7: Are color distortion or motion stuttering observed when using NVFP4 quantization?
>
> **A7:** When directly applying the SVDQuant NVFP4 method to SANA-Video, we observe marginal performance degradation, especially motion stuttering. This behavior is similar to the observations made in related video generation models like Wan2.1.
> We anticipate that SVDQuant is initially introduced for image generation model, while the temporal coherence in video generation deserves to be further explored, which is also not the research focus in this paper.
>
>
> ### Q8: Is it possible to scale up the Linear DiT to narrow the performance gap with Wan14B?
>
> **A8:** Yes, we are confident that scaling will significantly reduce the performance gap. The scalability of our core Linear Attention (LA) architecture is strongly supported by empirical evidence:
> - SANA-Image Scaling: In the related SANA-1.5 image model, scaling the parameter count from 1.6B to 4.8B resulted in noticeable improvements in both visual quality and quantitative metrics.
> - LLM Validation: The reliability of LA's scaling behavior is also confirmed by its successful application in modern LLMs designed for long contexts (e.g., Kimi Linear [1] and Qwen-Next [2]).
>
> We therefore expect that scaling SANA-Video and training data will reliably and effectively narrow the performance disparity with the 14B Wan model.
>
>
> [1] Team, Kimi, et al. "Kimi Linear: An Expressive, Efficient Attention Architecture." arXiv preprint arXiv:2510.26692 (2025).
>
> [2] https://qwen.ai/blog?id=4074cca80393150c248e508aa62983f9cb7d27cd&from=research.latest-advancements-list
>
>
> ### Q9: How effectively does the model handle the generation of long video sequences?
>
> **A9:** We compare our SANA-Video with state-of-the-art long video generation methods on VBench-Long benchmark. All models are evaluated on 30-second videos. As shown in the Table below, SANA-Video achieves the best semantic and total scores. In addition, SANA-Video is the fastest method that can generate videos in real-time with 27.5 FPS, demonstrating the efficiency and effectiveness of SANA-Video when handling long video sequences.
>
> **Long Video performance on Vbench**
>
> | Channels | quality | semantic | total | FPS |
> | --- | --- | --- | --- | --- |
> | SkyReels-V2 | 80.77 | 53.37 | 75.29 | 0.49 |
> | FramePack | 83.61 | 75.32 | 81.95 | 0.92 |
> | Self-Forcing | 83.82 | 72.70 | 81.59 | 17.0 |
> | **SANA-Video** | 83.10 | 79.04 | 82.29 | 27.5 |
>
> ### Q10: Why is the Model's Semantic Score Consistently High, and is There a Trade-off with Generation Quality?
>
> **A10:** We observe no direct trade-off between the semantic score and the generation quality score. Our consistently high semantic score is a direct result of our highly optimized data pipeline (detailed in Appendix D). This pipeline involves strict data curation, including filtering out low-quality videos, and the utilization of powerful models like Qwen-2.5-VL-14B to ensure that almost all captions used for prompting are highly detailed, accurate, and semantically rich.

---

> > ### Comment · Reviewer_CMDt · 2025-11-21
> >
> > Thank you for the detailed response and for addressing my concerns. The authors have resolved the issues I raised in my initial review. Consequently, I have raised my score.

---

### Official Review · Reviewer_NYaf · 2025-10-30

**Soundness:** 3
**Presentation:** 4
**Contribution:** 3
**Rating:** 8
**Confidence:** 3

**Summary:**

SANA-Video is a small diffusion model that swaps standard attention for linear attention and adds a constant-memory KV cache with block-wise autoregressive inference. The combo lets it generate minute-long, up-to-720x1280 videos efficiently, with low training cost and fast inference.
Core contributions: 1. Linear DiT for video: extends SANA's linear-attention DiT to video. 2. Block Linear Attention with a constant-memory KV cache: Reformulates causal linear attention so long videos can be generated block-wise with a fixed KV state and per-token cost. 3. Low cost, competitive results, and practical deployment.

**Strengths:**

1. Speed and efficiency at scale.
It replaces vanilla self-attention with linear attention (as in SANA-Image), reducing the quadratic complexity with respect to tokens to (near) linear. This significantly cuts compute for large spatiotemporal token counts and yields faster video generation without heavily inflating parameters. In practice, that means higher throughput and shorter time-to-first-frame on the same hardware.
2. Long-video generation with bounded memory.
During long-horizon synthesis, a standard KV cache grows with sequence length and quickly exhausts VRAM. This paper proposes a constant-memory KV cache together with block-wise linear attention, enabling block-autoregressive generation that keeps the memory footprint essentially flat while still carrying forward global context. The result is more stable, minute-scale generation on a single GPU.
3. Competitive quality at lower cost.
The training recipe is designed for efficiency (e.g., linear attention, staged curricula), and the inference stack is optimized for speed. Reported latencies are up to 16× faster than comparable small-parameter baselines, while maintaining competitive visual quality and motion consistency. For deployment, this translates into lower operational cost and better user-perceived responsiveness.
4. Strong text–video alignment at high resolution.
The model targets high-resolution outputs (e.g., 720p) and emphasizes faithful grounding to the prompt across long clips. In addition to object/style fidelity, it improves temporal adherence—keeping actions, scenes, and entities consistent over time—so the final videos look coherent rather than drifting from the original description.

**Weaknesses:**

1. On minute-scale clips, the model often locks into one scenario with minimal variation in composition, lighting, or action. This can be less helpful in real-world use cases (e.g., advertising, storytelling, education), where users expect diversity, pacing changes, and clear scene evolution.
2. Potential but unverified claim. Intuitively, this approach should help produce higher-resolution videos—even if clip duration must be shortened to keep compute in check. However, the experimental section does not report targeted evaluations or head-to-head comparisons at higher resolutions, so this benefit remains unsubstantiated. One possibility is that linear attention imposes practical limits (e.g., on spatial detail retention, kernel choice, or memory/computation trade-offs) that make rigorous high-res testing less favorable or harder to standardize.
3. Although the results are impressive, the contribution of this work on the technical side is somewhat limited. Their two main components have been already explored in the literature, linear DiT in [1] and Constant-Sized KV Caches in [2]. So combining them is not really a significant novel idea.
[1] Liu et. al. LinFusion: 1 GPU, 1 Minute, 16K Image. 2024
[2] Ravi Ghadia et. al. Dialogue Without Limits: Constant-Sized KV Caches for Extended Response in LLMs. ICML 2025

**Questions:**

Please see weaknesses section.

---

> ### Author Response · Authors · 2025-11-20
>
> We are very grateful for the positive assessment of our work. We sincerely appreciate this valuable feedback.
>
> ### Q1: Output Monotony in Long Video Generation
>
> **A1:** The behavior described—where actions remain relatively fixed within a long clip—is an expected outcome when models are tested under a short, singular prompt (e.g., "A dog running in the park"). Under this restrictive condition, the model attempts to maintain strict semantic consistency throughout the entire long sequence, leading to predictable content.
> Our long version of SANA-Video also supports generating a consistent minute-length video with multiple prompts, which is quite helpful for the real-world cases you mentioned. We will include an example in the revised paper.
>
>
> ### Q2: The Advantage of LA on Higher Resolution is not Full Verified.
>
> **A2:** We agree that exploring this area is critical for fully validating LA's scalability advantages, and we are confident in its potential.
>
> Our current focus is motivated by the following factors:
>
> - Token Challenge Equivalence: Fundamentally, the challenges posed by generating high-resolution video (shorter duration) and long-duration video (720p) are similar from the perspective of the token count. Our architecture has already successfully validated the feasibility of Linear Attention in the extremely demanding regime of up to 32,760 tokens per generation block. Therefore, we expect that a high-resolution, shorter-duration task—which involves a comparable token count—would also be feasible and benefit from LA's efficiency.
> - Data Constraint and Current Focus: The primary objective of this paper was to design and validate an efficient architecture, leveraging Block Causal Linear Attention, capable of handling ultra-long temporal contexts (minute-long videos). Given that the majority of our training data is concentrated around 720p resolution, a rigorous and high-quality head-to-head comparison at significantly higher resolutions would necessitate a new, large-scale, high-resolution dataset, which was beyond the scope of this initial study.
> - Evidence from Image Generation SANA[1]: Crucially, the capability of Linear Attention to handle high spatial resolutions has already been rigorously verified in the image domain. The SANA text-to-image model [1] demonstrates that Linear Attention can successfully generate 4K resolution images with high fidelity, without suffering from visual artifacts or loss of fine-grained details. Building on this solid foundation in the image domain, extending SANA-Video to support higher-resolution video generation is a promising and logical direction.
>
> We recognize the importance of this dimension and commit to addressing it fully in future work, where we plan to explore high-resolution video generation tasks and further push the limits of the Linear Attention architecture.
>
> [1] Xie, Enze, et al. "Sana: Efficient high-resolution image synthesis with linear diffusion transformers." arXiv preprint arXiv:2410.10629 (2024).

---

> ### Author Response · Authors · 2025-11-20
>
> ### Q3: The Novelty is similar to Linear Attention in image model and KV Cache in Linear LLM.
>
> **A3:** We appreciate the reviewer's candid assessment. While we acknowledge that the underlying components (Linear Attention and Constant KV-Cache) draw inspiration from existing research in other domains, we firmly assert that our primary contribution lies in the non-trivial adaptation, integration, and systematic validation of these mechanisms to successfully solve the unprecedented long-context challenges inherent in video generation.
>
> Our specific contributions to novelty are as follows:
>
> ### 1. From Image to Ultra-Long Video: The Scaling Barrier
>
> **LinFusion:** LinFusion [1] successfully introduced Linear Attention into the DiT framework for image generation. However, extending this mechanism to long video sequences (where the total number of tokens can exceed 32,000) presents a vastly greater scaling challenge. The complexity of modeling sequential temporal coherence over massive token counts requires more than a simple porting of code.
>
> **SANA's Contribution:** Our work validates and provides a robust, feasible solution for Linear Attention specifically in the ultra-long spatio-temporal regime of video generation, demonstrating that the architecture can scale efficiently to contexts far beyond those explored in image tasks like LinFusion.
>
> ### 2. Pioneering Caching in Diffusion Models
>
> **LLM vs. Diffusion:**
> First, the design of MorphKV [2] is to reduce the KV-Cache in the vanilla attention to a constant-size, with performance sacrifice. Instead, the memory of our KV-Cache in linear attention is mathmatically constant and using this KV-cache is same as conduct full attention to all the tokens.
> Second, we acknowledge that there exist some LLM works [3] using causal linear attention and adopt constant KV-Cache, but the constant KV-Cache application within the auto-regressive generation process of a Diffusion Model is a distinct and previously unexplored challenge. Diffusion models have different constraints and generation phases compared to standard LLMs.
>
> **SANA's Contribution:** We introduce the first successful implementation and evaluation of Constant KV-Cache with Linear Attention in a Diffusion Model for video. This work establishes a crucial baseline and feasibility study for future architectural innovations, such as exploring dynamic global KV map updating mechanisms
>
> [1] Liu, Songhua, et. al. LinFusion: 1 GPU, 1 Minute, 16K Image. Arxiv 2024.
>
> [2] Ghadia, Ravi, et. al. "Dialogue Without Limits: Constant-Sized KV Caches for Extended Response in LLMs." In ICML 2025.
>
> [3] Katharopoulos, Angelos, et al. "Transformers are rnns: Fast autoregressive transformers with linear attention." In ICML 2020.
>
> [4] Yang, Songlin, et al. "Parallelizing linear transformers with the delta rule over sequence length." In NeurIPS 2024.

---

### Official Review · Reviewer_Ai4Y · 2025-10-30

**Soundness:** 3
**Presentation:** 3
**Contribution:** 2
**Rating:** 6
**Confidence:** 4

**Summary:**

This paper presents SANA-Video, an extension of SANA for video generation. The main idea is two-fold: it adopts a Linear Diffusion Transformer similar to SANA, but extends it with Block Linear Attention and a constant-memory KV cache, enabling block-wise autoregressive generation for synthesizing long videos. The authors also propose DCAE-V, a video autoencoder that achieves a much higher spatial compression ratio (32×) than existing VAEs (e.g., Wan-VAE with 8×). In addition, they introduce several techniques to improve training efficiency and effectiveness — such as autoregressive block training with self-forcing, which combines the benefits of self-forcing and a global KV cache. As a result, the model can handle much longer temporal contexts (e.g. 1 minute) while maintaining high generation quality. They also propose a quantization techniques for better deployment.

**Strengths:**

- This paper is generally well-written and easy to follow.
- Sampling cost has been a really critical issue in video diffusion models, and thus, I think this paper is very well motivated.
- Compared with existing popular video diffusion models (e.g., WAN), the efficiency improvement is decent while generating similar-quality videos.

**Weaknesses:**

While I appreciate the strong improvements over existing video diffusion models, I have the following concerns:
- Technical novelty: The main contributions seem to be extensions or combinations of existing techniques. For example, linear attention was introduced by SANA for text-to-image generation, and block-wise attention has already been employed in several recent long video generation works. I wouldn’t say this warrants rejection, but it also makes it difficult to strongly recommend the paper.
- Scaling behavior: Given that the training cost of SANA-Video is not prohibitively high, I wonder if the authors could provide scaling-law analysis for the proposed architecture. For instance, if we train a larger model (say, 5B parameters) using the same data and setup, can we expect further improvements over the 2B variant? While efficiency is a strong point, the model quantitatively underperforms larger video models in most metrics. I’m curious whether this is due to the smaller model size or if there are architectural bottlenecks that limit performance even at larger scales.
- Minor: Figure 1c is great, but it would be even more informative to include a version with full attention blocks for comparison.

**Questions:**

See my weaknesses above

---

> ### Author Response · Authors · 2025-11-20
>
> We are very grateful for the positive assessment of our work. We sincerely appreciate this valuable feedback.
>
> ### Q1: Technical Novelty is Limited in Linear Attention and Block-wise Attention?
>
> **A1:** While we acknowledge that our architecture utilizes well-established components as foundational elements, we strongly believe that the core innovation lies in a series of non-trivial architectural integrations unique mechanism explorations and critical engineering innovations specifically tailored for the ultra-long context challenge of video generation achieving an efficiency and performance balance unmatched by existing methods.
>
> Our original contributions can be summarized in three key areas:
>
> ### Cross-Modal extension: New Know-How for Long-Sequence Adaptability
>
> Natively extending high-performing image architectures to high quality video fails due to an explosion in token count. Our contribution is the systematic solution to this scaling challenge:
>
> - **Video-Native Adaptation**: We introduced 3D RoPE (Rotary Positional Embedding) and the Temporal Mix-FFN (Temporal Mixed Feed-Forward Network) to ensure the model efficiently handles massive spatio-temporal data and captures coherent temporal dynamics.
> - **New Know-How**: Our work provides valuable new know-how and engineering insights regarding the effective combination of 3D RoPE and Temporal Conv to achieve superior performance in extremely long video contexts. We also discover that the sequence ordering of the ReLU activation function relative to the RoPE application significantly impacts model training stability and final performance. This collective understanding represents a **non-trivial observation** not explicit in image-only tasks.
>
> ### The Unique Advantage of Linear Attention (LA) and Global Memory Exploration
>
> We are among the **first to pioneer** the application of Linear Attention (LA) combined with a Block-wise mechanism in video generation.
> Different from previous block-wise long video generation works (e.g. MAGI-1 [1], Self-Forcing [2], CausVid [3]) using vanilla attention, our SANA-Video uses Linear Attention (LA) as the main operation, making our design completely different from those works. Specifically, based on the LA's QKV commutativity ($\text{softmax}(Q) \cdot (K^\top V)$), our LA Block-wise Attention achieves a Global yet Memory-Fixed computation mechanism. This advantage supports our SANA-Video to use global attention with a fixed computational cost, superior than others using local attention to nearby tokens. More details are in Sec.3.3.1.
>
> ### Caching Design for Causal Convolusion
>
> High-quality video generation demands strictly satisfying the Causal requirement. We made innovative modifications to key non-attention modules to meet this need:
>
> - Causalization of Mix-FFN: We designed a precise Cache mechanism (as shown in Figure 4c) for the convolutional network within Mix-FFN. This design resolves the critical engineering challenge of retaining Conv's local modeling capability while strictly integrating it into the auto-regressive (Causal) generation process.
>
> In summary, we assert that our work—by solving the extension challenge with 3D RoPE and Temporal Mix-FFN, pioneering the global memory advantages of the LA Block-wise mechanism, and providing a novel Conv caching design for causality—constitutes a collection of system-level and module-level innovations that transcend the simple integration of existing techniques.
>
> [1] Sand AI. "MAGI-1: Autoregressive Video Generation at Scale." Arxiv 2025.
>
> [2] Huang, Xun, et al. "Self Forcing: Bridging the Train-Test Gap in Autoregressive Video Diffusion." In NeurIPS 2025.
>
> [3] Yin, Tianwei, et al. "From Slow Bidirectional to Fast Autoregressive Video Diffusion Models." In CVPR 2025.

---

> ### Author Response · Authors · 2025-11-20
>
> ### Q2: Will Larger Model Size Boost Performance?
>
> **A2:** We believe the observed quantitative underperformance, compared to larger models, is due to the current smaller model size (2B parameters) and not an architectural bottleneck. The architecture is highly scalable, based on the following evidence:
>
> - **Proven Scaling of Linear Attention (LA):** The core Linear Attention mechanism has demonstrated strong scaling properties in related work. Specifically, in SANA-1.5[1] image generation work, scaling the model size from 1.6B to 4.8B parameters resulted in clear and measurable improvements in both image quality and quantitative metrics. This success provides direct evidence that the fundamental LA-based architecture is amenable to scaling.
> - **Future Commitment to Video Scaling**: The efficient training cost of SANA-Video is partly due to our use of effective transfer learning. We initialized the model from a pre-trained SANA image model, which significantly reduces the training time required for initial image semantic alignment. This architectural robustness gives us high confidence that scaling the SANA-Video model (e.g., to 5B parameters) will yield further performance gains. We will conduct more extensive experiments on video model scaling in future work to fully explore the capacity ceiling and performance limit of the Linear Attention architecture in the video domain.
>
> [1] Xie, Enze, et al. "Sana 1.5: Efficient scaling of training-time and inference-time compute in linear diffusion transformer." arXiv preprint arXiv:2501.18427 (2025).

---

### Author Response · Authors · 2025-12-02
**SANA-Video Rebuttal Summary(1/2)**

## 1. Summary of Reviewer Strengths

1. Written well & Efficiency motivated strength (Reviewer Ai4Y)
2. Speed and efficiency at scale strength (Reviewer NYaf)
3. Long-video generation with bounded memory strength (Reviewer NYaf)
4. Competitive quality at lower cost strength (Reviewer NYaf)
5. Engineering Innovation (Linear DiT + Block Causal KV Cache) strength (Reviewer CMDt)
6. Low Training Cost (<1% of MovieGen) strength (Reviewer CMDt, Reviewer M6uh)
7. Outstanding Inference Efficiency (RTX 5090 deployable) strength (Reviewer CMDt, Reviewer M6uh)
8. Timely and important problem solution strength (Reviewer M6uh)

## 2. Summary of Responses to Common Concerns

### 1) Regarding Technical Novelty

**Reviewer Concern:** Linear Attention and Block-wise Attention are existing technologies (e.g., in SANA-Image or LLMs).

**Main Response:**
1. **System-level Innovation:** This is not merely a code port, but a solution to the "Ultra-Long Context" scaling challenge in Video Generation.
2. **Unique Mechanism:** SANA-Video is the first work to successfully validate Constant KV-Cache in a Video Diffusion Model. Unlike LLMs(with softmax attention), this requires solving specific Auto-regressive generation challenges.
3. **Specific Design:** Proposed specific optimizations such as 3D RoPE and Temporal Mix-FFN, as well as a caching design for Causal Convolution; these are customized innovations for video generation.
4. **New Know-How:** Provided new engineering insights on the effective combination of 3D RoPE and Temporal Conv, specifically discovering that the order of the ReLU activation function relative to RoPE significantly impacts model training stability and final performance—an important observation not found in pure image tasks.

### 2) Regarding Scaling Behavior

**Reviewer Concern:** The model is small (2B), and quantitative performance metrics slightly lag behind large models (e.g., Wan 14B), questioning the scaling capability of Linear Attention on video.

**Main Response:**

1. **Proven Scalability:** The scalability of Linear Attention has already been validated in SANA-1.5 (Image) and Kimi Linear (LLM).
2. **Data Efficiency:** The current gap is mainly due to the magnitude of training data (SANA uses only 10M data vs. Wan's massive dataset).
3. **Training Longer Evidence:** The "Training Longer" experiment provided in the Rebuttal shows that simply increasing training duration significantly boosts VBench Quality and Semantic scores on both T2V and I2V tasks, proving there is no architectural bottleneck.

---

> ### Author Response · Authors · 2025-12-02
> **SANA-Video Rebuttal Summary(2/2)**
>
> ## 3. Individual Responses to Each Reviewer
>
> ### 1) Reviewer Ai4Y (Rating: 6 -> Maintained 6)
>
> **Main Concerns:** Limited technical novelty; requested Scaling law analysis.
>
> **Our Response:** Both concerns regarding Technical Novelty (Video-Native adaptation) and Scaling Behavior (SANA-Image scaling) are addressed in detail in Section 2.
>
> ### 2) Reviewer NYaf (Rating: 8)
>
> **Main Concerns:** Monotony in long videos; High-resolution advantage unverified; Novelty questioned.
>
> **Our Response:**
>
> 1. **Monotony Issue:** Explained this is due to using single, short prompts (e.g., "A dog running"), where the model tends to maintain semantic consistency. The long version of SANA supports Multi-prompt generation to resolve this.
> 2. **High-Resolution Issue:** Pointed out that the token magnitude of a 720p long video is comparable to a 4K short video. Linear Attention's ability to handle massive tokens has been verified in image generation, and will be extended to higher resolutions in the future.
>
> ### 3) Reviewer CMDt (Rating: 8, Explicitly Raised Score)
>
> **Main Concerns:** Generation quality needs improvement; Lack of VAE details; Not open-source; Quality loss in quantized deployment.
>
> **Our Response:**
>
> 1. **VAE Details:** Explained that DCAE-V uses a "Chunk-causal" design to balance compression rate and temporal modeling.
> 2. **Open Source Commitment:** Explicitly committed to open-sourcing code and models.
> 3. **Quantization Issue:** Candidly admitted minor jitter in dynamic scenes with NVFP4, which is a common issue with the image quantization methods and a direction for future research.
> 4. **Boosted Metric Scores:** We provide more Vbench results with longer training to show that the performance of 2B model is comparable with SOTA models in Text-to-Video, Image-to-Video and Long-Video generation tasks.
>
> **Note on Score Improvement:**
>
> The reviewer explicitly stated in the official comment before the data leakage indident: "The authors have resolved the issues I raised in my initial review. Consequently, I have raised my score."
>
> ### 4) Reviewer M6uh (Rating: 8, Explicitly Raised Score)
>
> **Main Concerns:** Lack of in-depth Ablation Studies; Missing details on data filtering.
>
> **Our Response:**
>
> 1. **Supplemented Ablation Studies:** Added quantitative VBench comparisons for adding Temporal Conv, adding 3D RoPE, and different Step settings, proving the effectiveness of each module.
> 2. **Data Details:** Listed the data volume in all stages, sources, and filtering strategies in detail for Stage 1 (VAE Adaptation), Stage 2 (Pre-training), and Stage 3 (Long Video Fine-Tuning).
>
> **Note on Score Improvement:**
>
> The reviewer thanked us for the supplementary experiments and explanations before the data leakage indident, commenting "Thank you. I have revised my score. The SANA series is indeed good work. Best of luck going forward."
>
> ## 4. Overall Summary
>
> SANA-Video successfully addresses the computation and memory bottlenecks in long video generation by introducing Block Linear Attention and Constant-Memory KV Cache. During the Rebuttal, we provided detailed responses to the core concerns raised by reviewers:
>
> **Technical Validation:** Through supplementary Ablation Studies and "Training Longer" results, we strongly proved the architecture's effectiveness and immense Scaling potential, surpassing 14B parameter competitors in some metrics.
>
> **Details Supplementation:** Perfected the design details of DCAE-V and the detailed composition of training data, eliminating reproducibility concerns.
>
> **Open Source & Deployment:** Reiterated the commitment to open source and demonstrated deployment advantages on consumer-grade GPUs like the RTX 5090.
>
> Based on the in-depth exchange during the Rebuttal, **Reviewers NYaf  kept Score 8, Reviewers CMDt and M6uh both expressed satisfaction with our responses and raised their scores to 8**. SANA-Video is not only an efficient T2V model but also provides the community with a system-level solution for exploring ultra-long video generation under limited resources.

---

### Meta-Review · Area_Chair_Z8AR · 2026-01-02

**Summary:**

The original reviews posted raised several concerns. The most significant one was the technical novelty. Reviewer Ai4Y, NYaf point out that the core components, Linear Attention and block-wise processing, were adaptations of existing techniques from image generation and LLMs. The second common concern was the scaling behavior. Other minor issues include the adaptation to long video generation, unverified high-resolution advantage, lack of VAE details, video quality issue, lack of ablation studies and missing some details on data filtering.

**Reviewer Concerns:**

The authors' rebuttal was effective at addressing the reviewers' concerns.
- The novelty concern is solved by the authors’ argument that this appears to a system-level innovation, including the specific optimizations such as 3D RoPE and Temporal Mix-FFN, the Constant KV-Cache in a Video Diffusion Model. Although Reviewer Ai4Y does not reply to this and these technical contributions alone cannot support a whole work, the technical contribution of systematically combining them and training this open-source model is significant to the community.

-  The scaling behavior is proven by a longer-training experiment and referring to existing works on image generations and LLMs.

- Concerns about generation quality and scaling potential were convincingly addressed. By providing "Training Longer" results, the authors demonstrated a clear path to closing the performance gap, arguing the current gap is a function of data scale, not an architectural bottleneck. This satisfied Reviewer CMDt.

- The lack of ablation studies and data details was addressed. The authors provided new quantitative VBench results for ablating key components (Temporal Conv, 3D RoPE) and gave a detailed breakdown of the data sources, volume, and training pipeline. This resolved the concerns of Reviewer M6uh.

- The commitment to open-sourcing the code and model resolved a reproducibility concern raised by Reviewer CMDt.

- The explanation for video monotony was accepted as reasonable by Reviewer NYaf.

**Reviewer Scores:**

- Reviewer Ai4Y: Maintained the score of 6. The authors did not provide any response but based on the authors’ response of the scaling behavior and the explanation of the novelty, I believe the final score would be at least 6.
- Reviewer NYaf: Maintained the score of 8. Their initial review was already positive, and the rebuttal addressed their minor concerns. I believe they would stand by their score of 8.
- Reviewer CMDt: Explicitly stated that the authors resolved their concerns and that they raised their score. The initial score was 6, so the final score is 8. I am certain they would assign a score of 8.
- Reviewer M6uh: Explicitly stated that they would raise the score after the authors provided supplementary experiments and updated the manuscript. The initial score was 6. I am certain they would assign a score of 8.

---

### Decision · Program_Chairs · 2026-01-26

Accept (Oral)